# Cellular and neurochemical basis of sleep stages in the thalamocortical network

**Giri P Krishnan[1]\*, Sylvain Chauvette[2,3], Isaac Shamie[4], Sara Soltani[2,3], Igor Timofeev[2,3], Sydney S Cash[5], Eric Halgren[4], Maxim Bazhenov[1]**

[1]Department of Medicine, University of California, San Diego, La Jolla, CA, United States; [2]Department of Psychiatry and Neuroscience, Université Laval, Québec, Canada; [3]Centre de Recherche de l'Institut Universitaire en Santé Mentale de Québec, Université Laval, Québec, Canada; [4]Departments of Radiology and Neurosciences, University of California, San Diego, La Jolla, CA, United States; [5]Department of Neurology, Massachusetts General Hospital and Harvard Medical School, Boston, United States

**Abstract** The link between the combined action of neuromodulators in the brain and global brain states remains a mystery. In this study, using biophysically realistic models of the thalamocortical network, we identified the critical intrinsic and synaptic mechanisms, associated with the putative action of acetylcholine (ACh), GABA and monoamines, which lead to transitions between primary brain vigilance states (waking, non-rapid eye movement sleep [NREM] and REM sleep) within an ultradian cycle. Using ECoG recordings from humans and LFP recordings from cats and mice, we found that during NREM sleep the power of spindle and delta oscillations is negatively correlated in humans and positively correlated in animal recordings. We explained this discrepancy by the differences in the relative level of ACh. Overall, our study revealed the critical intrinsic and synaptic mechanisms through which different neuromodulators acting in combination result in characteristic brain EEG rhythms and transitions between sleep stages.

**\*For correspondence:** gkrishnan@ucsd.edu

**Competing interests:** The authors declare that no competing interests exist.

## Introduction

During sleep, unique electrophysiological rhythms are observed in the EEG, intracellular recordings, EOG and EMG (*Steriade et al., 1993a*; *Niedermeyer and da Silva, 2005*). These features form the basis for classification of sleep into different stages: rapid eye movement sleep (REM), and stages N1 (Stage 1), N2 (Stage 2) and N3 (Stage 3) of non-REM sleep (*Rechtschaffen and Kales, 1968*; *Iber and Medicine, 2007*; *Silber et al., 2007*). A typical night of sleep consists of 4–5 sleep cycles of transitions across stages in healthy adults. Each sleep cycle shows progression of sleep stages in the following order, N1 → N2 → N3 → N2 → REM (*Feinberg and Floyd, 1979*). Approximately 75% of sleep is spent in NREM stages. There is also a greater amount of N3 (also termed Slow Wave Sleep, SWS) earlier in the night, whereas REM stage is relatively dominant during later cycles of sleep (*Aeschbach and Borbely, 1993*).

Each sleep stage is characterized by specific patterns of the brain electrical activity. The N1 stage consists of slow eye movements and low-amplitude low-frequency (4–7 Hz) EEG activity in humans. The sleep stage N2 is dominated by sleep spindle oscillations at 10–17 Hz (in humans) with waxing-and-waning field potentials. Spindles last about 0.5–2 s and recur every 2–20 s. Similar events have been studied in non-human mammals, where their frequency may be as low as 7 Hz. During the N3 delta band (0.2–4 Hz), rhythms are found in the EEG (*Rechtschaffen and Kales, 1968*; *Amzica and Steriade, 1998*). This stage is dominated by the slow oscillation, consisting of active (Up) and silent (Down) cortical states alternating at frequency 0.2–1 Hz, that are prominent and visible in the EEG,

**eLife digest** There are several stages of sleep that cycle repeatedly through the night with each producing distinctive patterns of electrical activity in the brain. It is thought that these patterns may help us to remember things that have happened throughout the day. Cells in parts of the brain called the hypothalamus and the brainstem control transitions between sleep stages. They regulate the release of chemicals known as neuromodulators in many parts of the brain, including the cortex and thalamus, which play the roles in memory and learning. Researchers now know how the neuromodulators influence the properties of individual brain cells. However, it is not clear how coordinated action of many neuromodulators result in the patterns of electrical activity seen in the brain during each stage of sleep.

Krishnan et al. used a computer model to investigate how three of these neuromodulators – acetylcholine, histamine and GABA – shift electrical activity in the brain between sleep stages. The computer model was able to recreate the network of brain cells in the cortex and thalamus and how this network responds to the changes in the levels of neuromodulators. The study found that simultaneous and balanced changes of acetylcholine, histamine, and GABA work together to shift the brain between the stages of sleep and to initiate patterns of the brain electrical activity specific to the different sleep stages.

Krishnan et al. predict that the relative differences in the level of acetylcholine in the brains of humans, cats and mice may explain why different species have different patterns of electrical activity during sleep. The study also found that an anesthetic drug called propofol may induce sleep-like patterns of electrical activity in the human brain by affecting the levels of all three of the neuromodulators. More studies are needed to look at how the networks of cells in the cortex and thalamus communicate with the brainstem, and how changes in the levels of neuromodulators affect memory and learning.

and in the extracellular and intracellular recordings (*Steriade et al., 1991*, *1993a*; *2001*, *Werth et al., 1996*; *Timofeev et al., 2000*, *2001*). During Up states, most cells within the cerebral cortex are relatively depolarized and may generate action potentials; during Down states, most cortical neurons are hyperpolarized and do not fire (*Steriade et al., 1993b*; *Contreras and Steriade, 1995*; *Timofeev et al., 2000*). The slow oscillation may nest faster spindles, which are commonly found during Down-to-Up state transitions (*Molle et al., 2002*; *Clemens et al., 2007*). During REM sleep, electrical brain activity *resembles* that of the awake state, including mixed signals with bursts of alpha (7–12 Hz) activity (*Cantero and Atienza, 2000*). Precise coordination of different EEG rhythms during sleep is believed to be critical for memory consolidation which manifests in reactivation of specific neural activity patterns – sleep replay – that have been observed in different brain areas including the hippocampus, amygdala, neocortex and striatum (*Nadasdy et al., 1999*; *Pennartz et al., 2004*; *Euston et al., 2007*; *Popa et al., 2010*; *Bendor and Wilson, 2012*).

Neuromodulators, such as acetylcholine (ACh), GABA, histamine (HA), serotonin (5-HT) and norepinephrine (NE), are known to vary significantly during sleep and awake as well as across sleep stages. Compared to the awake state, in N2 and N3, the levels of ACh and monoamines such as HA and NE are reduced while the level of the inhibitory neurotransmitter GABA is increased (*Aston-Jones and Bloom, 1981*); (*McCormick, 1992*; *Vazquez and Baghdoyan, 2001*; *Lena et al., 2005*; *Vanini et al., 2011*). During REM sleep, the level of ACh is increased, but monoamines and GABA are reduced (*Baghdoyan and Lydic, 2012*). While intracellular and synaptic targets of specific neuromodulators are somewhat known, we still lack a clear understanding of how the orchestrated action of many neuromodulators leads to the very specific types of the brain electrical activity in awake and sleep.

In this study, we present a comprehensive computational model of the thalamocortical system implementing effects of neuromodulators and identify the critical intrinsic and synaptic neuronal mechanisms required to explain transitions between sleep stages within an ultradian cycle. We predict that the differences in the temporal dynamics of spindle and delta band oscillations observed in the LFP recordings of cats and mice and ECoG activity of human subjects during SWS (N3 in

humans) could arise from the relative differences in the level of neuromodulators. We further apply the model to explain electrical activity observed in propofol-induced anesthesia. Our study predicts that even minor changes of neuromodulators may affect the properties of sleep spindles and sleep slow oscillation in NREM sleep, thus possibly affecting the dynamics of memory consolidation.

## Results

Levels of the ACh, HA and GABA vary across sleep stages in vivo. During NREM stages (N2 and N3), the ACh and HA levels are reduced compared to awake; during REM, the ACh and HA levels are increased (*Baghdoyan and Lydic, 2012*) (*Figure 1B*). The level of the inhibitory neurotransmitter, GABA, is increased during N2 and N3 compared to awake, while reduced during REM sleep (*McCormick, 1992*; *Vanini et al., 2011*). To link the action of these neuromodulators to characteristic brain electrical activity in different sleep stages, we developed a thalamocortical network model implementing the basic effects of the ACh, HA and GABA (see 'Materials and methods'). The model included populations of pyramidal (PY) cells and inhibitory interneurons (IN) in the cortex, and thalamic relay (TC) and reticular nucleus (RE) neurons in the thalamus (*Figure 1A*); it was designed based on our previous studies of sleep slow oscillations and sleep spindles (*Bazhenov et al., 1998*, *1999*; *Timofeev et al., 2000*; *Bonjean et al., 2011*; *Chen et al., 2012*; *Wei et al., 2016*).

Specific effects of neuromodulators were implemented by changing the strength of the intrinsic and synaptic currents in the cortical and thalamic neurons (see 'Materials and methods' for more details). While our model does not capture the entire spectrum of changes associated with known effects of neuromodulators, we identified a minimal set sufficient to account for characteristic changes of brain electrical activity across the sleep-wake cycle. Briefly, a reduction of ACh was implemented as an increase in the conductance of the $K^+$ leak current in TC, PY and IN neurons and a reduction of the $K^+$ leak current in RE cells (*McCormick, 1992*) (*Figure 1C,D*). The level of ACh also determined the maximal conductance of the excitatory AMPA connections (*Figure 1C,D*), so that reduction of ACh led to an increase in AMPA connection strength, in agreement with experimental findings (*Gil et al., 1997*; *Kimura et al., 1999*; *Hsieh et al., 2000*). The effect of HA was implemented as a shift in the activation curve of a hyperpolarization-activated cation current, $I_h$, in TC cells (*Figure 1B*) (*McCormick and Williamson, 1991*; *McCormick, 1992a*). Higher values of HA led to a positive shift in the $I_h$ activation curve (*Figure 1B*) as seen in vitro (*McCormick and Williamson, 1991*). The level of HA also determined the strength of the $K^+$ leak conductance in all neurons (see 'Materials and methods' section). The level of GABA determined the maximal conductance of the GABAergic synaptic connections. We also tested in the model the effect of extracellular GABA concentration on tonic inhibition.

Below we will first show that combined action of all these factors is sufficient to explain transitions between the awake state, the N2 and N3 stages and the REM sleep state. Next, we map the relationship between the space of electrical brain activities during sleep and the neurochemical space. We then discuss the impact of specific neuromodulators on brain electrical activity and sleep patterns. Next, we apply our model to explain discrepancies in the animal and human data regarding phase coupling between specific sleep rhythms. Finally, we apply our model to simulate the effect of propofol anesthesia on brain electrical activity.

### Collective action of acetylcholine, histamine and GABA explains transitions between sleep and awake states

When the model implemented the intracellular and synaptic changes associated with the putative actions of ACh, HA and GABA, it produced the main features of the different sleep stages observed in human and animal studies. During the awake period, electrical activity in the cortical PY and IN and thalamic RE cells was sparse, that is mean firing was around 2–3 Hz and 4–5 Hz in PY and IN cells, respectively, while the TC neurons primarily remained silent and RE cells were busting intermittently (*Figure 2A* and *Figure 3A*). To test the stability of network dynamics, we applied brief thalamic stimulation in the form of a short (100 ms) DC pulse during the awake state. It triggered a transient increase of firing in cortical PY neurons and INs, after which the network returned to the baseline, as observed in experiments with a sensory input during alert awake conditions (*Bruno and Sakmann, 2006*; *Hengen et al., 2016*). The average activity (simulated LFP) during awake period

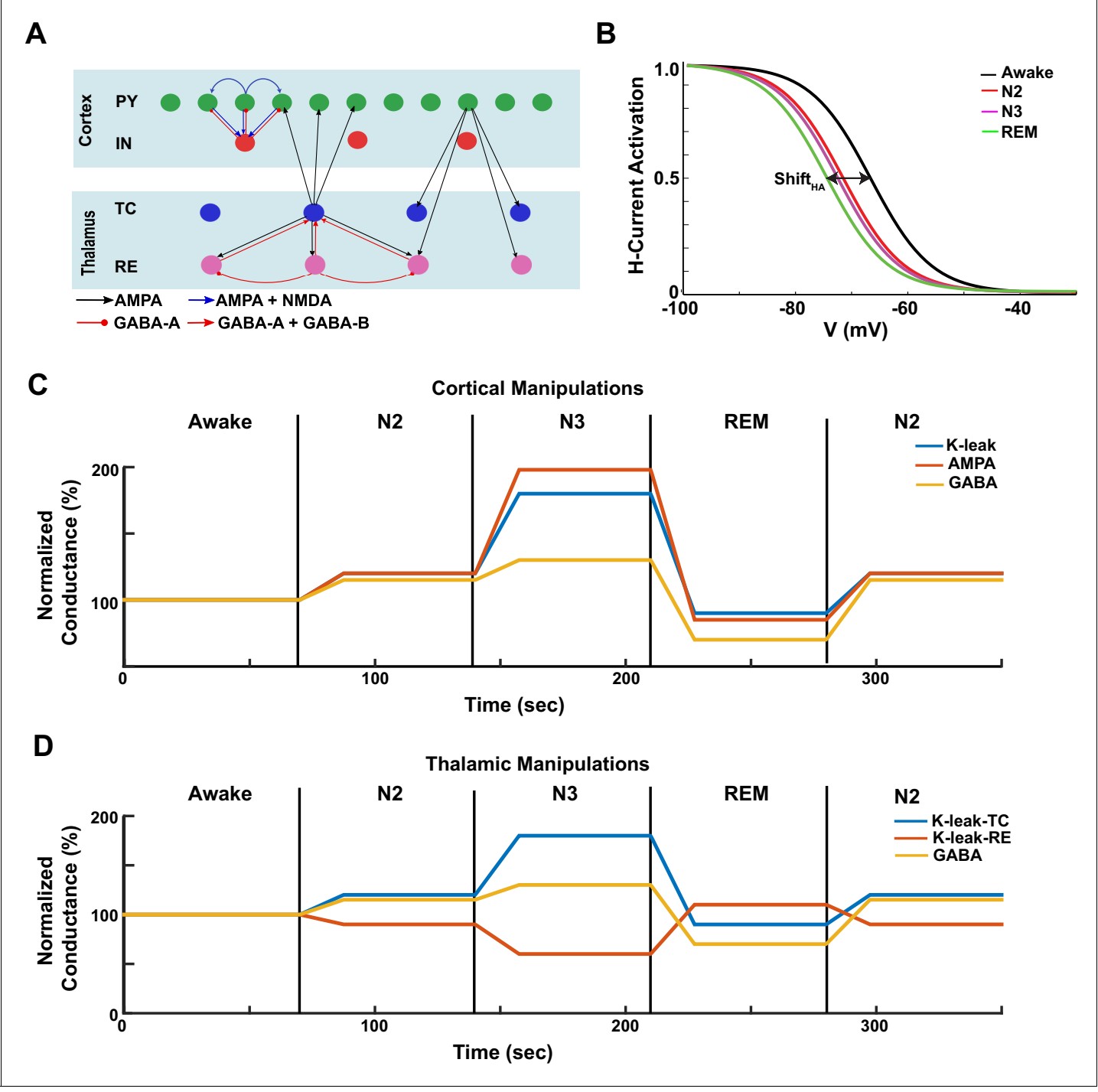

**Figure. 1.** Model design. (**A**) Network structure of the model. PY - cortical principal neurons, IN - cortical inhibitory interneurons, TC - thalamocortical (relay) neurons, RE - thalamic reticular neurons. There were 500 PY, 100 IN, 100 RE and 100 TC cells in the network. (**B**) The activation function for the h-current under different HA levels, corresponding to the different awake and sleep states as indicated in the legend. $\text{Shift}_{HA} = 0$ for the REM state. (**C**) A sequence of changes to the maximal conductances for AMPA, GABA and $K^+$ leak currents in the cortex to model sleep stage transitions. (**D**) A sequence of changes in $K^+$ leak currents in thalamic cells, and the maximal conductance for GABA synaptic currents within the thalamus.

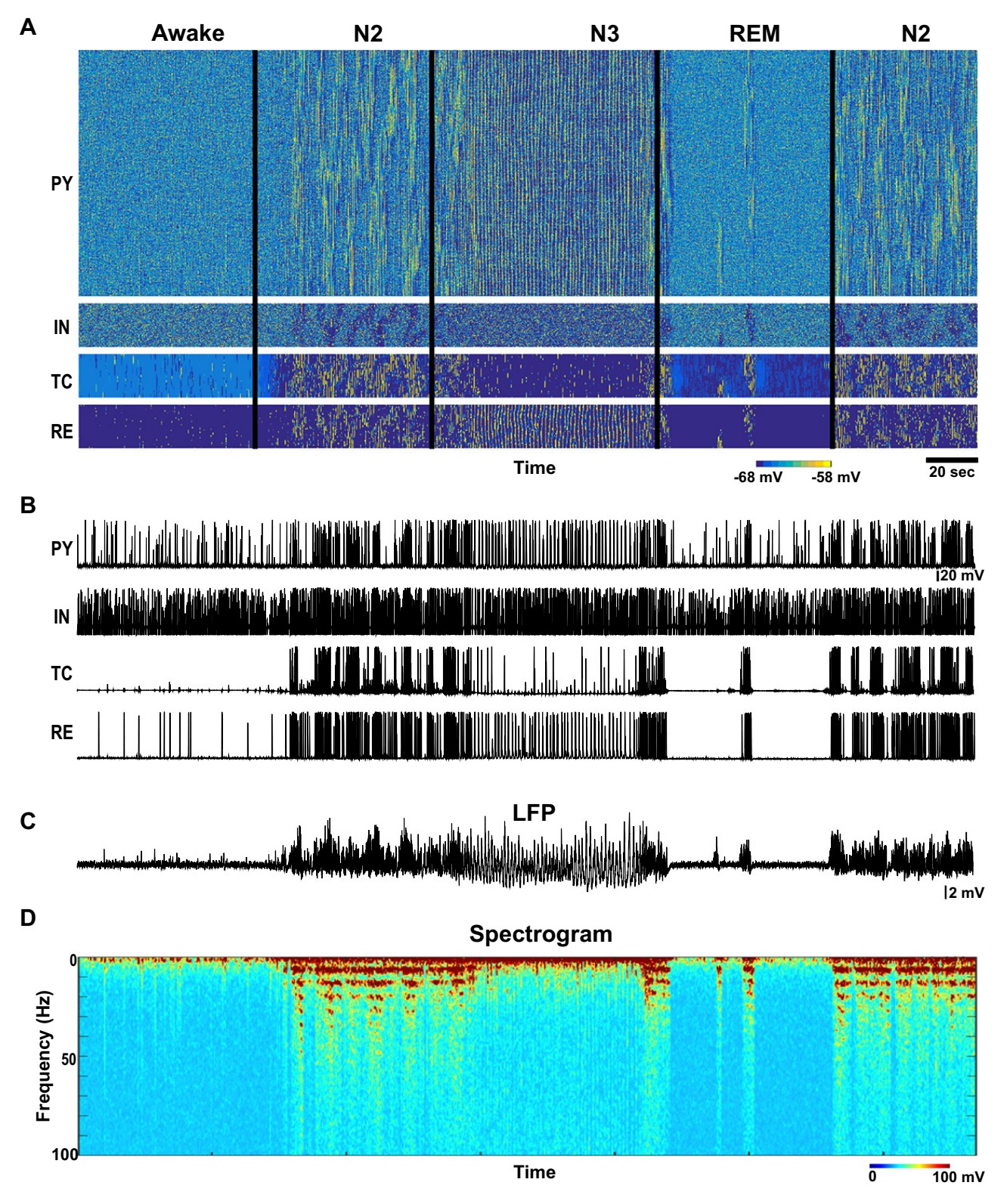

**Figure 2.** Network activity in awake, N2, N3 and REM sleep stages. (**A**) Top: Activity of all classes of neurons in the thalamocortical network model (500 PY, 100 IN, 100 RE and 100 TC cells) with the neuromodulators (ACh, GABA and HA) varying as shown in the *Figure 1B–D*. X-axis is time and Y-axis is neuron index. Colors indicate membrane voltage. For awake, N2, N3 and REM stages, respectively, ACh was 100%, 80%, 50% and 115%; HA was 100%, 40%, 30% and 10%; and GABA was 100%, 115% 130% and 75%. (**B**) Membrane voltages of the representative selected neurons from the network. (**C**)

*Figure 2 continued*

LFP calculated from PY network (mean voltage of all 500 PY neurons). Note spindles and slow oscillations during N2 and N3, respectably. (D) Spectrogram (based on the sliding time window FFT) of the LFP shows activity in spindle frequency (8–15 Hz) during N2, rare alpha-burst events (8–15 Hz) during REM, and slow oscillation (0.5–1 Hz) during N3.

(*Figure 2C*) showed no oscillatory activity or large deflections from baseline, reflecting desynchronized neuronal firing.

The N2 stage was characterized in the model by the periods of spindles – thalamically-organized bursts of 7–15 Hz oscillations lasting 0.5 to 2 s each and recurring every 3–20 s (*Steriade et al., 1993a*). In the model, we observed a transition to N2-like activity following reduction of ACh and HA, and increase of GABA. No external stimulation was applied to initiate or maintain spindles; however, thalamic stimulation could trigger a spindle response. In this state of the model (see N2 in *Figure 2*), periods of 7–15 Hz oscillatory activity reappeared spontaneously and lasted for at least 2–5 s as revealed by the spectrogram (*Figure 2D*). Membrane voltages were hyperpolarized in the PY, IN and TC cells due to the reduction of the ACh (*Figures 2B* and *3B*), similar to the intracellular recording (*Steriade et al., 1993c*). In agreement with the prior intracellular data (*Contreras and Steriade, 1996*) and computational models (*Bazhenov et al., 2000*; *Bonjean et al., 2011*), spindles recurred every 3–10 s.

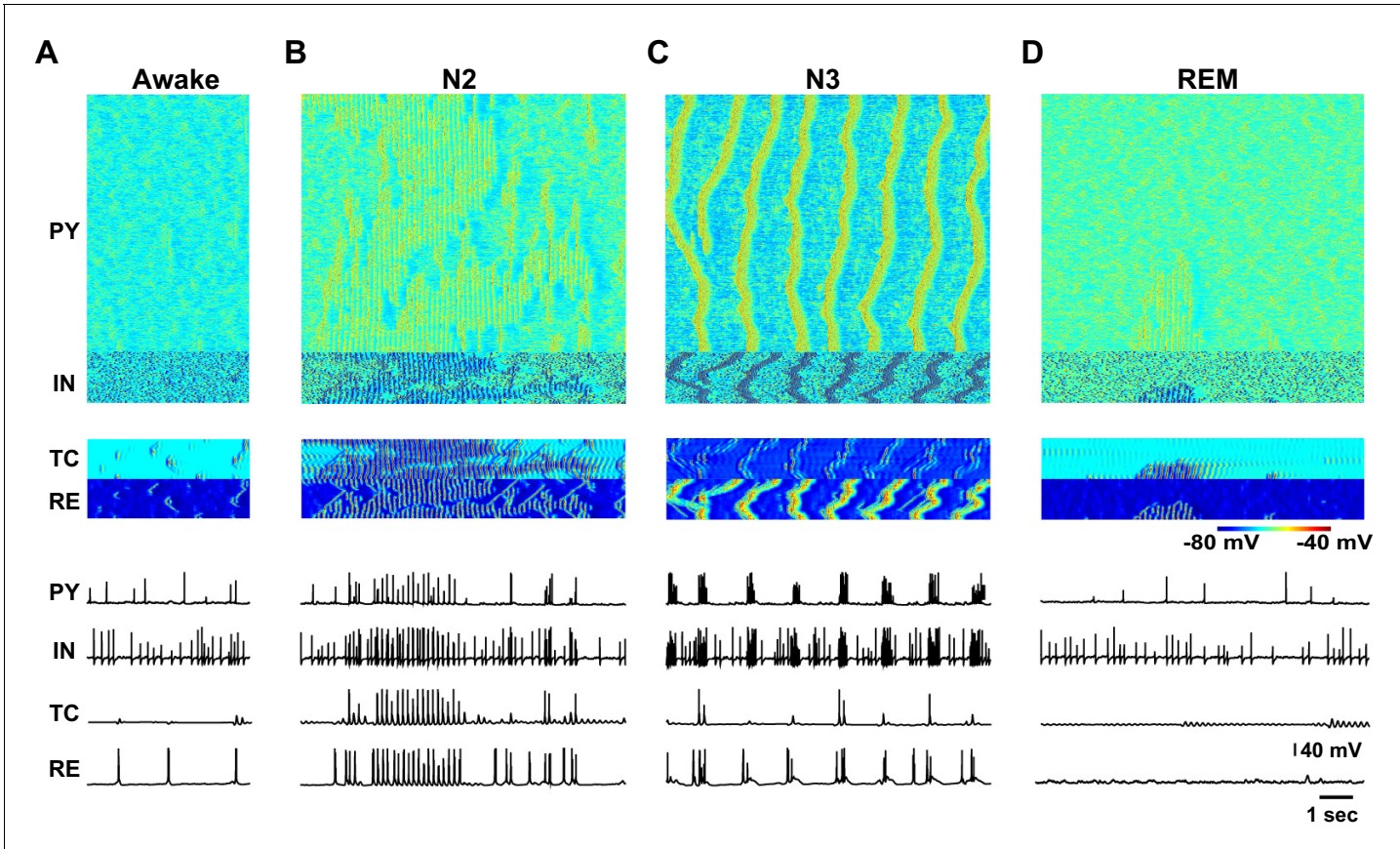

**Figure 3.** Characteristic patterns of electrical activity during different sleep stages. (**A**) Top: Network activity in PY, IN, TC and RE cells shown for a 5 s time window during awake. Y-axis is neuron index. Colors indicate membrane voltage. Bottom: membrane voltages of selected neurons from the network (neuron #250 in PY, #50 in IN, RE and TC populations). (**B–D**) Network activity during N2 (**B**) N3 (**C**) and REM (**D**) sleep for a 10 s time window. Legend is same as for panel A. Note a single localized in space 'alpha-burst' event in D.

When the levels of ACh and HA were reduced and GABA was increased compared to the simulations of the N2 state, N3-like activity appeared in the network (*Figures 2* and *3C*). In this state, a stereotypical slow oscillation consisting of transitions between Up and Down cortical states at a frequency of around 0.5–1 Hz was observed in the LFP, similar to experimental data (*Steriade et al., 1993c*). Up states were characterized by the higher spiking activity in all cortical pyramidal cells, interneurons and thalamic reticular neurons, while TC cells showed a depolarized state (similar to intracellular recordings from higher order thalamic nuclei [*Sheroziya and Timofeev, 2014*]) with a few spikes usually at transition from the silent to the active state; the Down states corresponded to the periods of the network silence (*Timofeev et al., 2000*; *Bazhenov et al., 2002*; *Compte et al., 2003*). As in vivo, faster spindle oscillations were found during Up states of the slow oscillation in the model.

To model REM sleep, the level of ACh was increased slightly compared to the awake period, whereas the levels of HA and GABA were reduced. In this network state, there was an overall increase in the cortical activity with very brief periods of 7–15 Hz oscillations that appeared on the background of desynchronized low-frequency cortical firing. The clusters of synchronized firing were somewhat similar to that observed during spindle oscillations. However, they appeared to be more localized in space. In the model, these brief periods of 7–15 Hz synchronized firing (*Figure 3D*) depended on the reduced level of monoamines in the thalamus and higher excitability in the cortex. We propose that these periods of synchronized oscillations correspond to the alpha or mu rhythms (7–13 Hz activity) that have been reported during REM sleep (*Cantero and Atienza, 2000*).

## Effect of neuromodulators on spindle and slow oscillation power and network synchronization

In order to reveal the specific role of different neuromodulators on network dynamics, the levels of ACh, GABA and HA were varied across a wide range (ACh: 0 to 120%, GABA: 25–225%, HA: 34 to 100% of the awake stage). Power was measured in the spindle (7–15 Hz) and delta (0.5–4 Hz) frequency bands, during 10 s periods for each combination of the neuromodulators. In our model, during N3 the delta band activity was dominated by slow oscillations (0.5–1 Hz). Synchronization among network sites was measured using the phase locking value (PLV) in a broadband frequency (0.5–20 Hz) range, between the LFPs obtained by averaging membrane voltages within five groups of neurons, each comprising 100 PY neurons. *Figure 4A* shows the summary of the results from all simulations projected to the spindle vs. delta power plane. Several clusters became apparent. Using cluster analysis (Gaussian mixture model) with spindle and delta power and PLV as inputs, 10 different clusters were identified (Akaike information criteria saturated around 10 components). The mean values of the clusters are indicated by the different color spheres in *Figure 4A* and examples of characteristic activity in each cluster are shown in *Figure 4C*.

We found that identified cluster states corresponded to either well-defined characteristic types of sleep activity (e.g. sleep spindles or slow oscillations) or various mixed states. When spindle and delta power were both low (e.g. clusters 3 and 4), the activity resembled an awake or REM sleep. In many instances of activity in cluster 4 (*Figure 4C*), we observed very brief and spatially localized periods of 7–15 Hz oscillations, which were similar to the alpha or mu rhythms recorded during REM sleep. For higher levels of delta power, two groups of clusters were observed: clusters corresponding to low spindle power (e.g. cluster 2) and clusters where spindle power was positively correlated to the delta power (e.g. clusters 5, 6). Cluster 2 with low-spindle power but high-delta power corresponded to the stereotypic transitions between Up and Down states (*Figure 4C*) as observed during the sleep slow oscillation in vivo, while clusters with a correlated spindle and delta power (clusters 5 and 6) corresponded to the mixed states, where brief (cluster 6) or long (cluster 5) periods of spindles appeared between periods of slow oscillation. Finally, a distinct cluster 1 corresponded to low-delta power, but high-spindle power. This cluster represented quasi-continuous spindle activity (*Figure 4C*).

The clusters shown in the *Figure 4A* were identified based on the quantitative characteristics of the electrical activity in the network. In order to link them to specific levels of the neuromodulators, all detected clusters were projected to the neurochemical space. Such projection was performed by fitting an ellipsoid around the means of each cluster in the three-dimensional space of the neuromodulators (*Figure 4B*). Thus, location of each ellipsoid provides a guide to the range of neuromodulators that produced a specific electrographic pattern. A multivariate ANOVA on the location of the

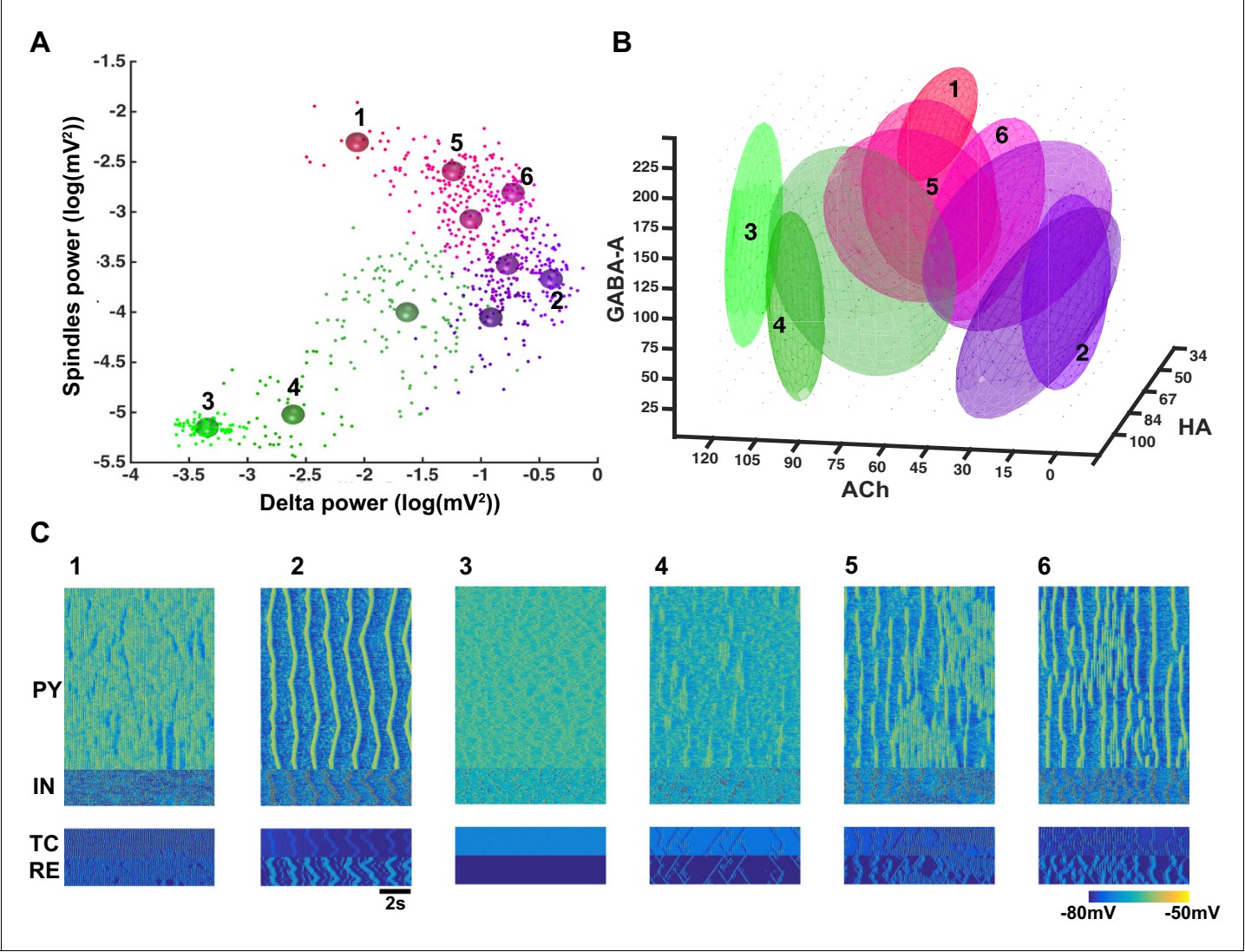

**Figure 4.** Cluster analysis of the electrophysiological and neuromodulatory spaces. (**A**) Delta (0.5–4 Hz) power of the global LFP (mean voltage of the 500 PY neurons) is plotted against spindle (7–15 Hz) power for different combinations of ACh, GABA and HA. Different colors represent different clusters identified based on the cluster analysis (mixed Gaussian model-based clustering was computed with spindle and delta power and Phase Locking Value (PLV) as inputs; 10 different clusters were identified). The PLV was computed for all pairs taken from 5 LFPs (mean voltage time series that were obtained by averaging every 100 PY neurons). Color spheres indicate the mean values of the clusters. (**B**) Projections of the clusters identified in panel A to the neuromodulator space of the ACh, HA and GABA. Each ellipsoid represents the approximate location of the clusters identified in **A**. (**C**) Characteristic electrical activity of the thalamocortical network (top to bottom: PY, IN, TC and RE cells) for each cluster identified in panels **A** and **B**. Cluster index is indicated above each plot and corresponds to the index in the panels **A** and **B**.

different clusters in the neuromodulator space was significant ($p < 10e{-}6$), suggesting that the clusters occupy different regions. Clusters 3 and 4 (green ellipsoids in *Figure 4B* corresponding to the awake or REM sleep like electrical activity) were associated with relatively high ACh levels such as that observed during awake and REM periods. Cluster 2 (violet ellipsoid in *Figure 4B* corresponding to the slow oscillation) was observed in the low ACh conditions, as found during N3. Clusters 5 and 6 (sleep stage 2 based on electrical activity) corresponded to intermediate levels of the neuromodulators. Finally, cluster 1 was observed for lower HA and higher GABA levels, in agreement with the data showing that strong GABA promotes spindle activity (*Parrino and Terzano, 1996*). In sum, our analysis confirmed a specific role of neuromodulators in controlling the electrical activity of the

thalamocortical network in different sleep and awake conditions, in agreement with past experimental studies (*Baghdoyan and Lydic, 2012*; *McCormick, 1992*).

## Minimal models of sleep stage transitions

In order to further identify the critical mechanisms responsible for state transitions in the thalamocortical network, we tested the minimal changes of the neuromodulators that were sufficient to generate the essential electrographic characteristics of each sleep stage (*Figure 5*). Changes in the neuromodulators were restricted to either thalamus or cortex to isolate the role of different regions. Furthermore, we focused on manipulations of the ACh and HA and not GABA, since in our model GABA modulation (within the physiological range) alone was not able to induce any state transitions. We found that in order to induce a transition from awake to N2, a reduction of HA in the thalamus was necessary but produced very focal and scattered spindles (*Figure 5C*). Reduction of both ACh and HA in the thalamus (*Figure 5D*) resulted in spindle activity that was similar to that in a full model (*Figure 5A*, see also *Figure 3B*). Cortical changes alone did not lead to spindles (*Figure 5B*) but were required to obtain cortical membrane potentials in the physiological range during N2. We concluded that reduction of ACh and HA in the thalamus, along with reduction of ACh in the cortex, constitute a minimal model needed to simulate the main features of the transition from the awake state to the N2 sleep state.

Transition from N2 to N3 could be observed when ACh was reduced in the cortex alone (*Figure 5F*). In contrast, even when all the other proposed neuromodulatory changes (reduction of ACh and HA, and increase of GABA) were applied in the thalamus, the thalamocortical network did not display N3-like activity (*Figure 5E*), suggesting that appearance of the slow oscillation during N3 requires neuromodulatory action within the cortex itself. Since reduction of the ACh has two main actions in our model, an increase in excitatory connection strength and an increase of leak currents, we examined if either of these changes alone could result in transition to the N3. Increasing excitatory AMPA conductance alone led to an activated state (*Figure 5G*), and an increase in the $K^+$ leak conductance alone led to a hyperpolarized quiescent state in the network (*Figure 5H*). Thus, both actions together were required for transition between the N2 and N3.

## Comparison between sleep stages in cats, mice and humans

We next examined if our thalamocortical model implementing varying levels of neuromodulators can explain LFP data from regions of cat and mouse neocortex across sleep stages (*Figure 6 A,B*). Surprisingly, visual scoring of cat LFPs revealed that there were only relatively short periods of the S2 (equivalent to N2; to describe animal experiments we use here terminology common in the animal studies) compared to the SWS (equivalent to N3). This result was consistent across animals. The power in the spindle (8–15 Hz) and delta (0.2–4 Hz) frequency bands was measured across all channels and the average power was plotted in *Figure 6C and D*, separately for the cortical and thalamic (VPL) recordings. The amplitude of the slow oscillation and spindles were both high during the S2 and SWS, and were reduced in awake and REM. This suggests that the SWS in naturally sleeping cats is marked with the relatively high-spindle activity. Slow (0.2–1 Hz) oscillations provided the major contribution to delta band activity during SWS. To further examine the interaction between sleep spindles and slow oscillations, we plotted the power in the spindle vs delta frequency bands across different cortical LFP electrodes (*Figure 6E*). We found a strong correlation between slow oscillation and spindle activity. In mice (*Figure 6F,G*), a similar relationship between spindle and delta power was observed across recordings from five animals (3 channels per mouse); a positive correlation was observed in majority of the channels (9 channels out of 15 channels showed significant correlation of which 7 were positively correlated [r ranged from 0.17 to 0.43; p<0.05] and 2 were negatively correlated [r was −0.41 and −0.43; p<0.05]). Recordings from four out of five mice revealed significant correlations. In one mouse (*Figure 6F–G*), frontal and posterior somatosensory electrodes showed a significant positive correlation between slow oscillations and spindle activity.

Our analysis of the neuromodulator space (*Figure 4*) suggests that a moderate reduction of ACh would lead to a network state with correlated variations in spindle and delta power (clusters 4, 5 and 6 in *Figure 4A*). Thus, in order to replicate our animal data (cat and mouse LFP activity), a computational model with a relatively high level of ACh during SWS (reduction to 75% of the baseline awake level) was required (*Figure 7A*). In contrast, in our control simulations of SWS (N3), the ACh was

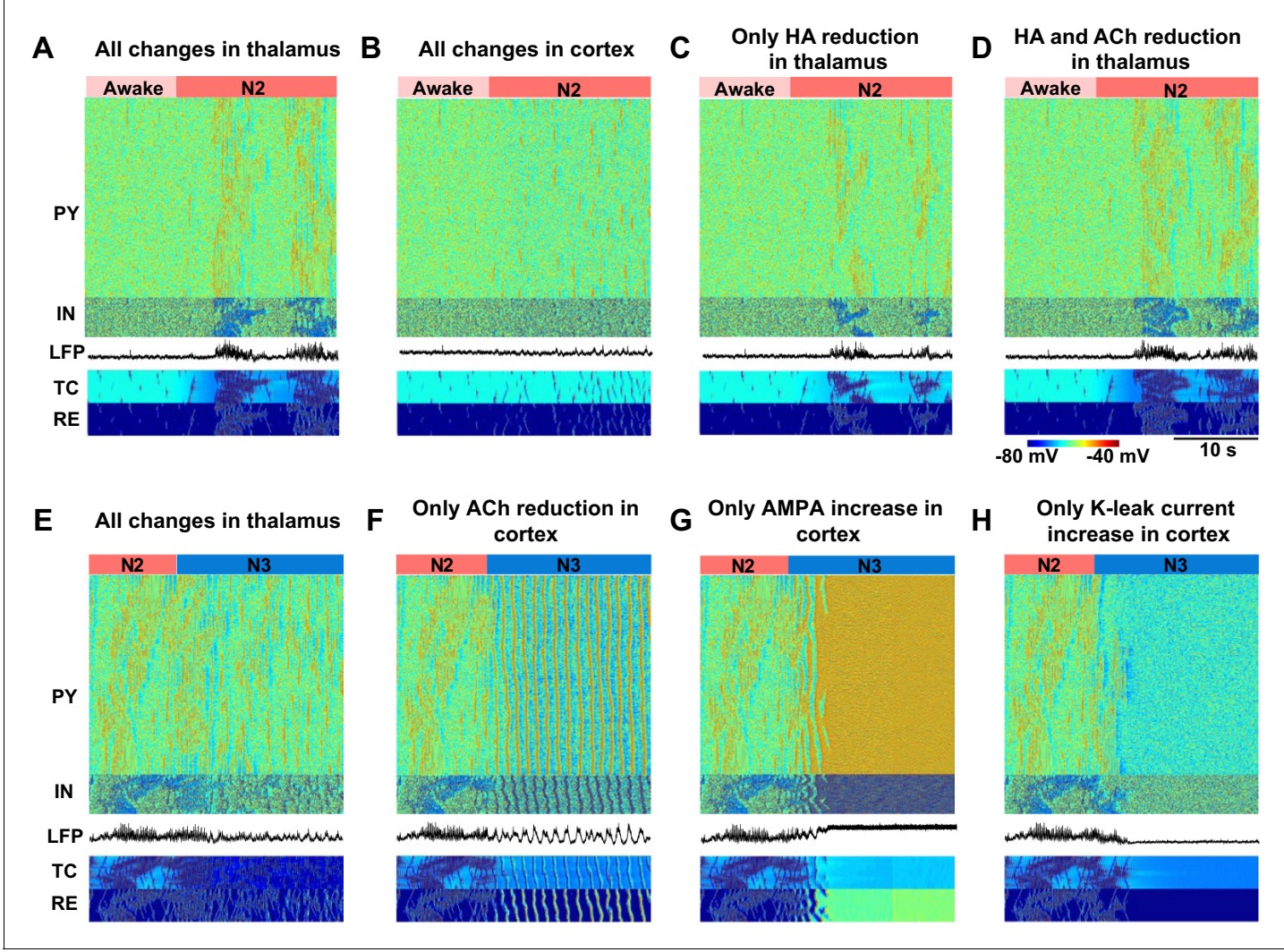

**Figure 5.** Minimal models of sleep state transitions. In all these simulations only specific indicated manipulations were performed. Other model parameters remained unchanged compared to the awake state. The time of the parameter changes is indicated by the bars at the top of each plot. Each plot shows activity of all classes of neurons in the thalamocortical network (500 PY, 100 IN, 100 RE and 100 TC cells) and the LFP. (**A**) All previously indicated manipulations corresponding to the awake to N2 transition (reduction of ACh and HA, and an increase in GABA) were performed in the thalamic network; no cortical changes. For awake stage: $ACh_{TC}$, $ACh_{RE}$, $Shift_{HA}$, $L_{GABA}$ (RE-RE, RE-TC) were 1.0, 1.0, −8.0 and 1.0 respectively. For N2 stage: $ACh_{TC}$, $ACh_{RE}$, $Shift_{HA}$, $L_{GABA}$ (RE-RE, RE-TC) were 1.25, 1.25, −3.0 and 1.15 respectively; $L_{ACh}$, $ACh_{PY}$ and $L_{GABA}$(IN-PY) were all fixed at 1.0. (**B**) All previously indicated manipulations corresponding to the awake to N2 transition (reduction of ACh and increase in GABA) were performed in the cortical network; no thalamic changes. For awake stage: $L_{ACh}$, $ACh_{PY}$, $Shift_{HA}$, $L_{GABA}$(IN-CX) were 1.0, 1.0, -8.0 and 1.0 respectively. For N2 stage: $L_{ACh}$, $ACh_{PY}$, $Shift_{HA}$ and $L_{GABA}$(IN-CX) were 1.25, 1.25, -3.0 and 1.15 respectively; $ACh_{TC}$, $ACh_{RE}$ and $L_{GABA}$ (RE-RE, RE-TC) were fixed at 1.0. (**C**) HA level was reduced in the thalamic network; no any other changes were performed. For awake and N2: only $Shift_{HA}$ was changed from −8 to −3; all other variables were fixed at awake level. (**D**) HA and ACh levels were reduced in the thalamic network; no any other changes were performed. For awake stage: $ACh_{TC}$, $ACh_{RE}$, $Shift_{HA}$, $L_{GABA}$ (RE-RE, RE-TC) were 1.0, 1,0, −8.0 and 1.0 respectively. For N2 stage: $ACh_{TC}$, $ACh_{RE}$, $Shift_{HA}$, $L_{GABA}$ (RE-RE, RE-TC) were 1.25, 1.25, −3.0 and 1.15 respectively; $L_{ACh}$, $ACh_{PY}$, $L_{GABA}$(IN-CX) were fixed at 1.0. (**E**) All specific manipulations corresponding to the N2 to SWS transition (ACh, HA and GABA levels modified) were performed in the thalamic network; no changes in the cortex. For N2 stage: $ACh_{TC}$, $ACh_{RE}$, $Shift_{HA}$, $L_{GABA}$ (RE-RE, RE-TC) were 1.25, 1.25, − 3.0 and 1.15 respectively. For N3 stage: $ACh_{TC}$, $ACh_{RE}$, $Shift_{HA}$, $L_{GABA}$ (RE-RE, RE-TC) were 2.0, 2.0, −8.0 and 1.3, respectively; $L_{ACh}$, $ACh_{PY}$ and $L_{GABA}$(IN-PY) were fixed at N2 stage level. (**F**) ACh level was reduced in the cortex; no any other changes were performed. For N2 and N3 stages, $L_{ACh}$ was 1.0 and 1.25; $ACh_{PY}$ was 1.0 and 1.25; all other variables were fixed at N2 level. (**G**) Only AMPA strength was increased in the cortex. For N2 and N3 stages, $L_{ACh}$ was 1 and 1.25, correspondingly; all other variables were fixed at N2 level. (**H**) K[+] leak current was decreased in the PY neurons. For N2 and N3 stages, $ACh_{PY}$ was 1 and 1.25, correspodingly; all other variables were fixed at N2 level.

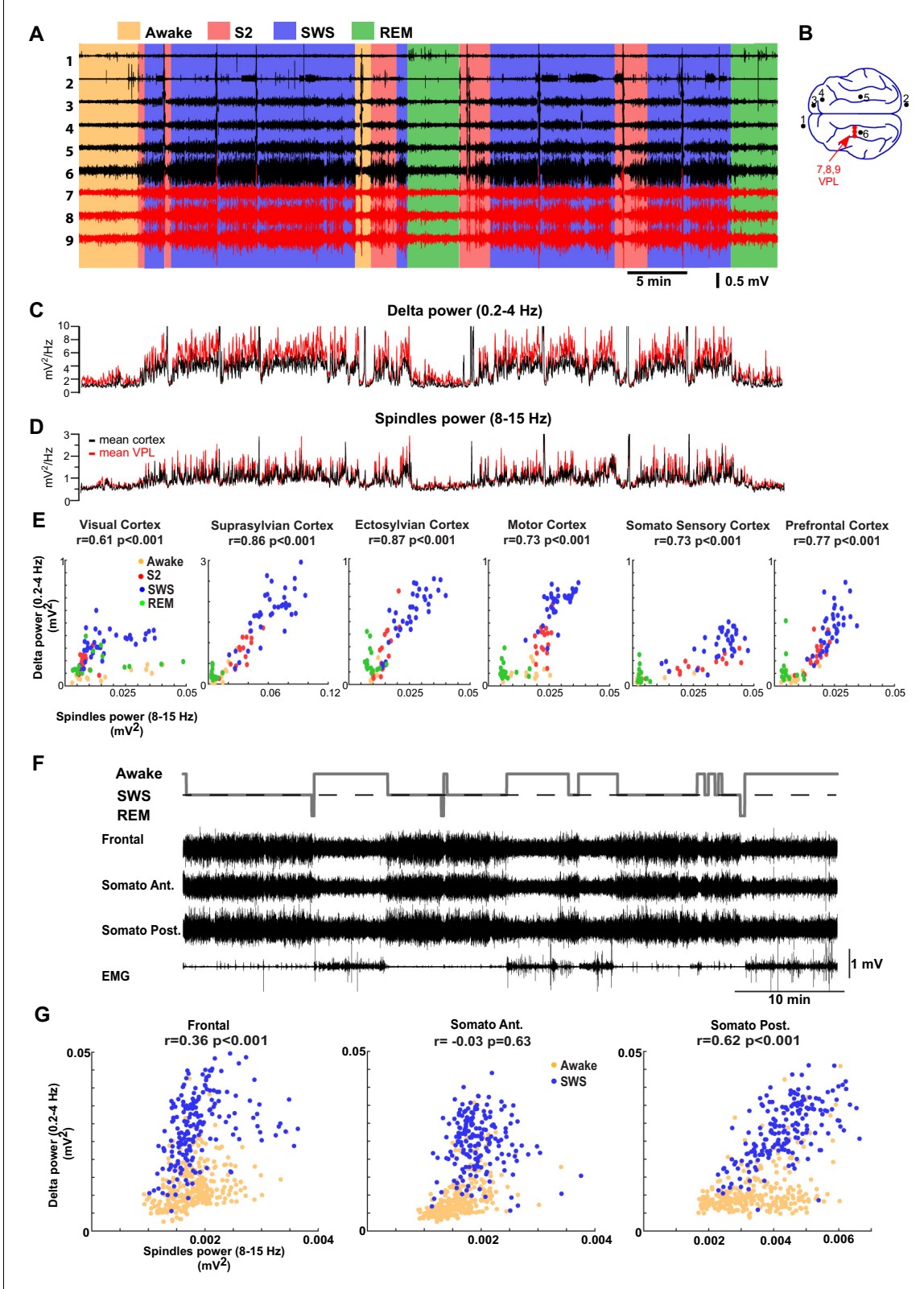

**Figure 6.** LFP recordings from cats and mice show positively correlated spindle and delta power during NREM sleep. (**A**). In vivo LFP recordings from different locations in the cortex (black lines) and the thalamus (red lines) of a non-anesthetized cat. (**B**) Locations of the recording electrodes. (**C,D**) Power at delta frequency (0.2–4 Hz) (**C**) and spindle frequency (8–15 Hz) (**D**) measured from 5 s time windows (1 s sliding window). (**E**) Spindle power is plotted against delta power for different channels. PCA was applied prior to computing the power and the principal components correlating strongly to

*Figure 6 continued*

EOG were removed; channel data were recomputed from remaining components to remove eye movement artifacts. Each dot represents the power in spindle and delta bands measured from a 30-s period of cat recordings. Note a positive correlation between delta and spindle power when combining S2 and SWS epochs. (**F**) Top: Hypnogram of recordings in mice. Bottom: In vivo LFP recordings from different cortical locations in mice. Note lack of S2 periods in mice. (**G**) Spindle power is plotted against delta power for different channels in mice. Each dot represents the power in spindle and delta bands measured from a 30-s window of data. There is a positive correlation between delta and spindle power during SWS epochs.

reduced to 57% of the baseline (*Figures 2* and *3*). Using the new model with the higher level of the ACh during SWS, we found a relative increase in the spindle activity during the SWS (*Figure 7A*) and

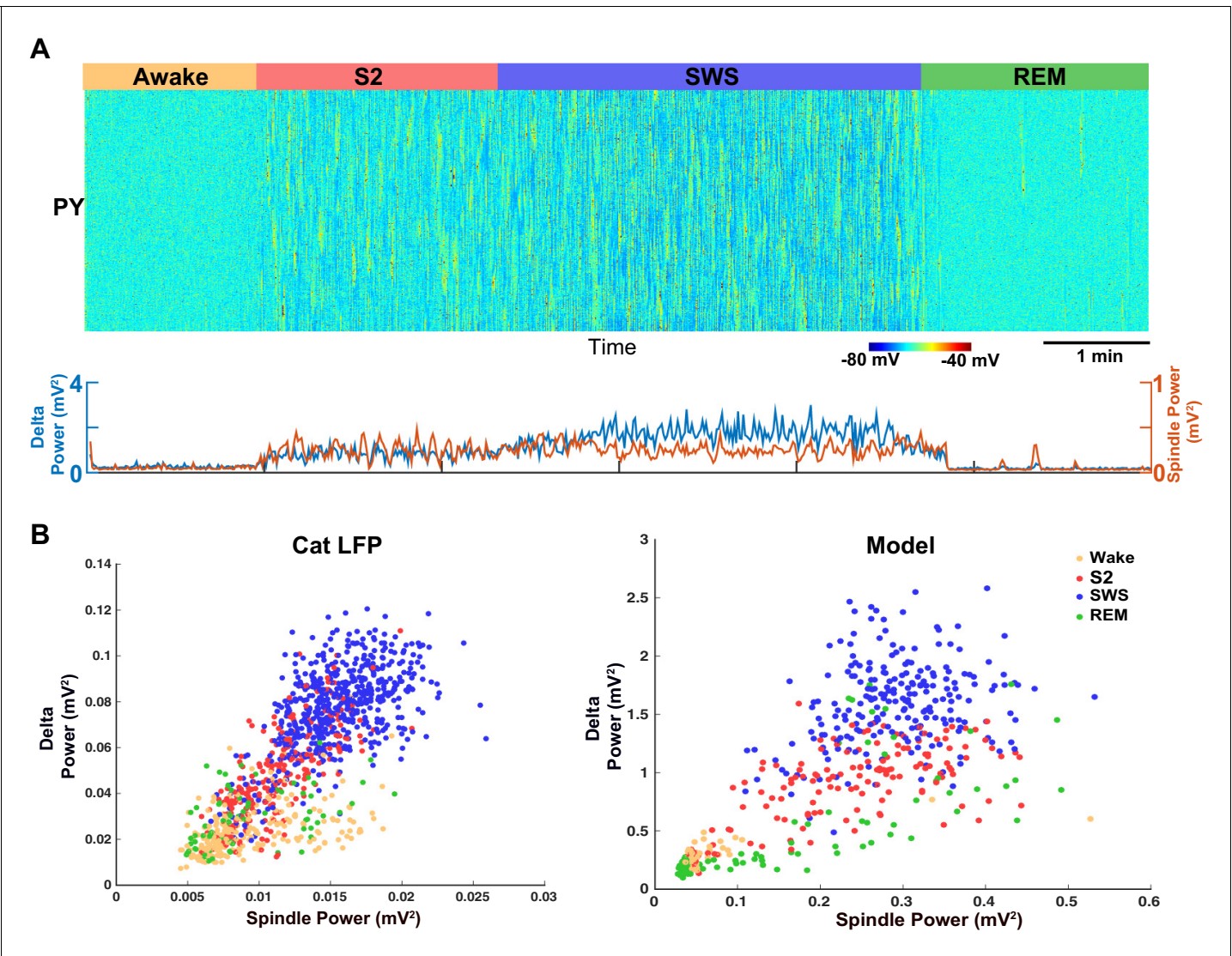

**Figure 7.** Model predicts temporal characteristics of the LFP activity in cats and mice. (**A**) The network model implementing moderate ACh reduction (to 75% of the baseline awake level) reproduces sleep stage transitions with pattern of activity similar to that recorded from cats and mice. (**B**) Spindle power (7–15 Hz) averaged across all electrodes is plotted against delta band power (0.5–4 Hz) for different sleep stages. PCA was applied to remove eye movement artifacts in cat recordings. Right: Spindle power vs. delta power based on the model simulations. Average activity within groups of 100 neurons was used as an estimate of the LFP. Each dot represents the power in spindle and delta bands measured from 2 s windows of data obtained from the model simulations.

a strong positive correlation between slow oscillations and spindle activity (*Figure 7B*, right), as was observed in the animal LFP data (*Figure 7B*, left).

Next, ECoG recordings from humans during different sleep stages were examined for the relationship between delta (0.01–2 Hz) and spindle (9–17 Hz) power (*Figure 8A,B*). (Note, these slightly different frequency bands are conventional in human studies.) In contrast to the pattern found in cats and mice, there was a significant *negative* correlation between the spindle and the delta power observed in successive 30 s epochs obtained during the N2 and N3 (*Figure 8C and D*) in humans. Generally, the slow oscillation was higher during N3 than during N2, whereas spindle activity was higher in N2 than in N3, resulting in a negative correlation across epochs. The average Pearson's correlation coefficient across 42 channels was −0.2595 (sd = 0.2281) (*Figure 8E*). Fifteen out of 42 electrodes showed a significant negative correlation (p<0.001), and no electrodes showed a significant positive correlation. We also observed periods of increased power in the 7–15 Hz frequency band during REM and awake suggesting transient periods of alpha (or mu) rhythm. To simulate the negative correlation between sleep slow oscillation and spindle activities, we again applied results from our analysis of the neuromodulator space (*Figure 4*). When the ACh level was reduced to 45% of the baseline during the N3 (compared to 57% used in the control model, see *Figures 2*, *3*), the network activity revealed a negative correlation between the delta and spindle frequency bands (*Figure 8 F and G*). These findings suggest that a relatively low level of the ACh during the N3 could possibly explain the negative correlation between the slow oscillation and spindle activities as observed in human ECoG recordings.

## Modeling the effects of propofol anesthesia requires the combined action of ACh, HA and GABA

We next applied our model to explain the features of the electrical activity induced by propofol anesthesia. In humans, propofol has shown to increase 8–15 Hz oscillations frontally and to reduce alpha oscillations posteriorly (*Murphy et al., 2011*; *Purdon et al., 2013*). Propofol application also increased the slow oscillations in the delta (1–4 Hz) frequency range across all regions. The onset of the loss of consciousness was closely matched by the increase of slow (<1 Hz) and 8–15 Hz oscillations (*Purdon et al., 2013*). We examined if our model could capture these changes in the EEG due to the action of propofol.

Propofol is a GABA agonist and is known to increase the decay time constant of the GABA-A mediated inhibitory post-synaptic potential (*Kitamura et al., 2003*). Multiple lines of evidence suggest that propofol also acts by reducing the action of ACh and HA. Indeed, propofol is shown to decrease the level of ACh in frontal cortex (*Kikuchi et al., 1998*) and to attenuate ACh receptor responses (*Flood et al., 1997*; *Murasaki et al., 2003*). Increasing ACh transmission prevents the action of propofol in humans (*Meuret et al., 2000*). Propofol inhibits the activity of the tuberomammillary nucleus (*Nelson et al., 2002*), although its action on ventrolateral preoptic nucleus neurons (*Liu et al., 2013*) leading to reduction of HA.

When these effects of ACh and HA were implemented in our model, as described before, and the decay time constant of GABA was increased, we observed a large increase in slow oscillations as well as oscillatory activity in 8–15 Hz frequency band (*Figure 9A*). In comparison to the model of natural sleep (*Figure 9C*), there was elevated 8–15 Hz power in the propofol condition (*Figure 9D*). This is consistent with our recordings from cats (not shown), where we found that 8–15 Hz power under propofol was high compared to natural sleep or ketamine-xylazine anesthesia. These results are also consistent with the previous computational models of propofol application, where the increase in decay time constant of IPSPs led to increase in 8–15 Hz oscillations (*Ching et al., 2010*). However, in our model, when only the time constant of the GABA-A IPSP was increased compare to the awake condition, the model failed to generate slow oscillations or the 8–15 Hz synchronized activity (*Figure 9B*). Overall, our study suggests that the known action of propofol in vivo may require its indirect effect on both the ACh and HA neuromodulatory systems.

## Role of GABA in sleep stage transitions

Experimental studies suggest that the concentration of extracellular GABA in the neocortex is higher during NREM sleep and it is reduced during REM sleep compared to the awake state (*Vanini et al., 2012*). Increase of extracellular GABA may reflect an increase of synaptic inhibition. Indeed, during

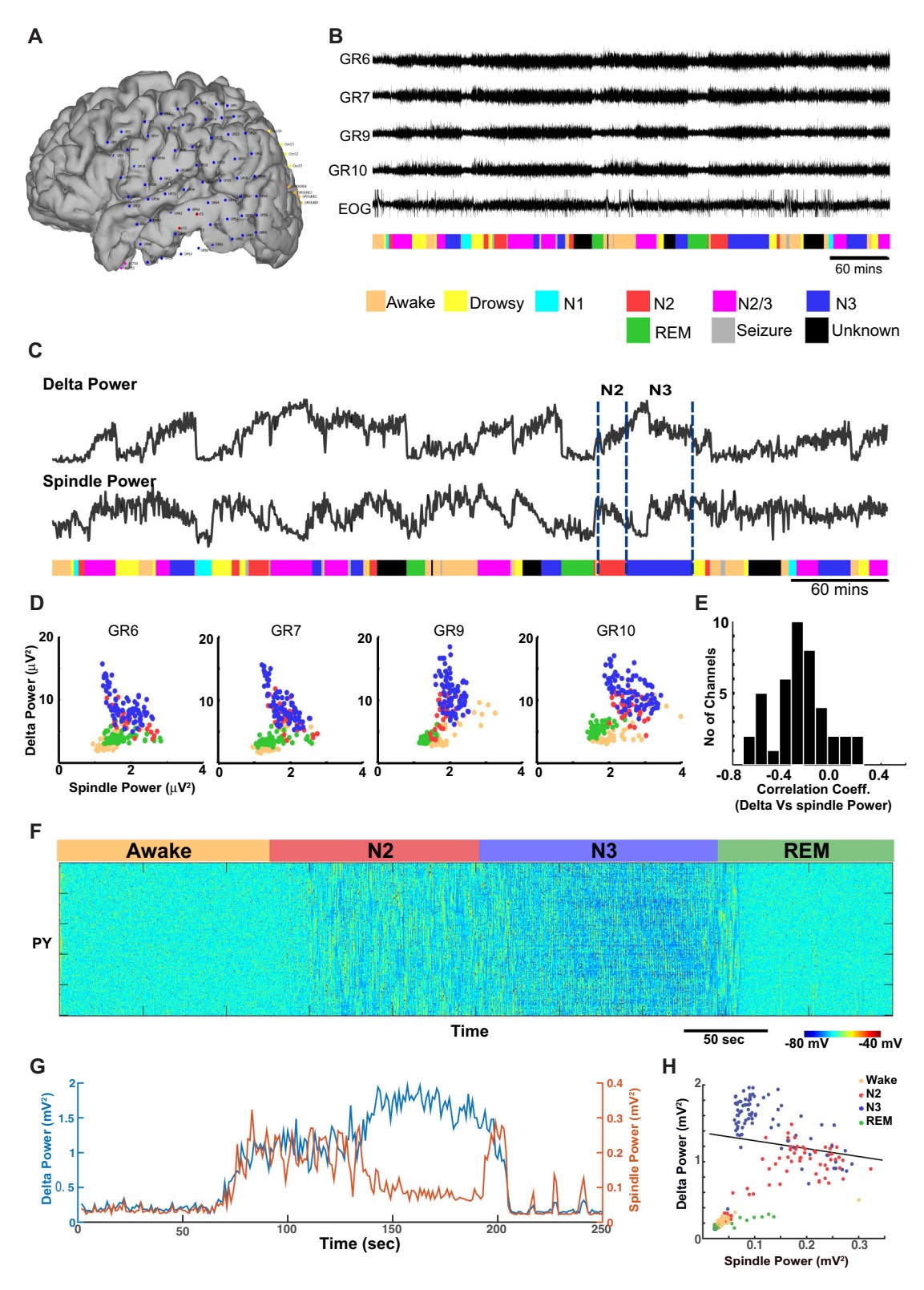

**Figure 8.** ECoG recordings from a human patient show negatively correlated spindle and delta power during NREM sleep. (**A**) Patient MG29 had a 64-contact grid and 4 strips implanted over the left hemisphere. After rejection of electrodes that either had poor contact or showed significant epileptic interictal activity, 42 electrode contacts remained for analysis. (**B**) Recording of patient for 9 hr beginning in the evening, during which patient was both awake and asleep. Shown are LFP over four grid electrodes, along with the right EOG channel. Beneath that is the sleep staging done in 30 s epochs.
*Figure 8 continued on next page*

*Figure 8 continued*

Staging was scored using information from scalp, EOG and intracranial electrodes. Data from Awake, REM, N2 and N3 were used for further analysis. A significant decrease in overall LFP amplitude was seen in REM and occasionally during awake. (C) Delta and spindle power over the same period. A Fast Fourier Transform with a window size of 30 s was done for delta (band 0.01–2 Hz) and spindle (9–17 Hz) band frequencies to obtain their power. All 42 electrodes were averaged and are plotted here. (D) Delta vs. spindle bands for 30 s time epochs were plotted for individual electrodes and separated based on sleep staging. There is a negative correlation between delta (dominated by slow oscillation) and spindle activities for these electrodes when combining N2 and N3 epochs. (E) A histogram of Pearson's R correlation coefficients between delta and spindle power during N2 and N3 across all 42 electrodes. (F) The computational model implementing significant ACh reduction (to 45% of the baseline awake level) reproduces sleep stage transitions with pattern of activity similar to that recorded from humans. (G) Spectral analysis of model activity. Power at delta frequency (0.2–4 Hz, red) and spindle frequency (8–15 Hz, blue) were measured from FFT obtained with non-overlapping sliding 2 s windows, similar to the analysis shown in C of actual recordings. (H) Inverse correlation of delta and spindle power in the model during combined N2 and N3 activity, similar to human recordings in D.

slow wave sleep, synaptic excitation and inhibition are balanced (*Rudolph et al., 2007*; *Haider et al., 2006*). Since decrease of ACh during stage N2/N3 sleep is known to increase excitatory connections (*Gil et al., 1997*), synaptic inhibition and thus phasic GABA release may be also increased, which would then be reflected in elevated extracellular GABA. The effect of increase of synaptic inhibition was implemented in our baseline model as reported above.

Nevertheless, the origin of the change in the extracellular GABA is still not fully understood. Increase of extracellular GABA may increase tonic inhibition but not necessarily be associated with increase of the phasic GABA release. Further, several studies revealed co-release of GABA with ACh and HA (*Saunders et al., 2015*; *Yu et al., 2015*) which suggests a decrease of GABA during NREM sleep while the observations from microdialysis experiments (*Vanini et al., 2012*) suggest an increase of GABA during NREM sleep. Therefore, in our study, we also tested models with (a) no change in the level of the GABA release; (b) increase of tonic inhibition reflecting increase of the extracellular GABA, based on the observations from microdialysis experiments (*Vanini et al., 2012*); and (c) tonic inhibition proportional to the ACh and HA levels, based on the evidence of the co-release of GABA with ACh (*Saunders et al., 2015*) and HA (*Yu et al., 2015*). In all these conditions (*Figure 1*), we found transitions between sleep stages similar to those reported in the baseline model. However, the model implementing co-release of GABA with ACh had poorly formed spindle activity during N2 and elevated alpha activity during REM sleep. Future experiments are required to distinguish between changes in tonic and phasic inhibition to match specific features of the synchronized oscillatory activity across sleep stages.

## Discussion

The functional state and the patterns of electrical activity of human and animal brain are continuously influenced by a broad range of neuromodulators (*McCormick, 1992*; *Steriade et al., 2001*; *Baghdoyan and Lydic, 2012*). Changes in the level of neuromodulators have been correlated with sleep induction as well as transitions between characteristic EEG sleep patterns (*McCormick, 1992*; *McCormick and Bal, 1997*; *Steriade et al., 2001*). However, the mechanisms whereby specific cellular and synaptic changes triggered by the combined action of neuromodulators result in transitions between sleep stages and their precise electrical characteristics remain poorly understood. In this new study, we used biophysically realistic computational models of the thalamocortical system to identify the minimal set of cellular and network level changes, linked to the putative action of the neuromodulators, that is sufficient to explain the characteristic neuronal dynamics during sleep as well as the transitions between primary sleep stages and from sleep to awake state. Our study predicts a critical role of the neuromodulators in controlling the precise spatio-temporal characteristics of neuronal synchronization that manifest in major sleep rhythms – sleep spindles and slow oscillations. Our results are consistent with in vivo intracellular and LFP recordings from animals and ECoG data from human subjects, as reported in this study.

Previous studies identified intrinsic and synaptic mechanisms involved in generating specific types of sleep rhythms. The survival of the slow oscillation (<1 Hz global electrical activity found during deep (N3) sleep) after extensive thalamic lesions in vivo (*Steriade et al., 1993b*), its existence in cortical slice preparations (*Sanchez-Vives and McCormick, 2000*), and the absence of the slow

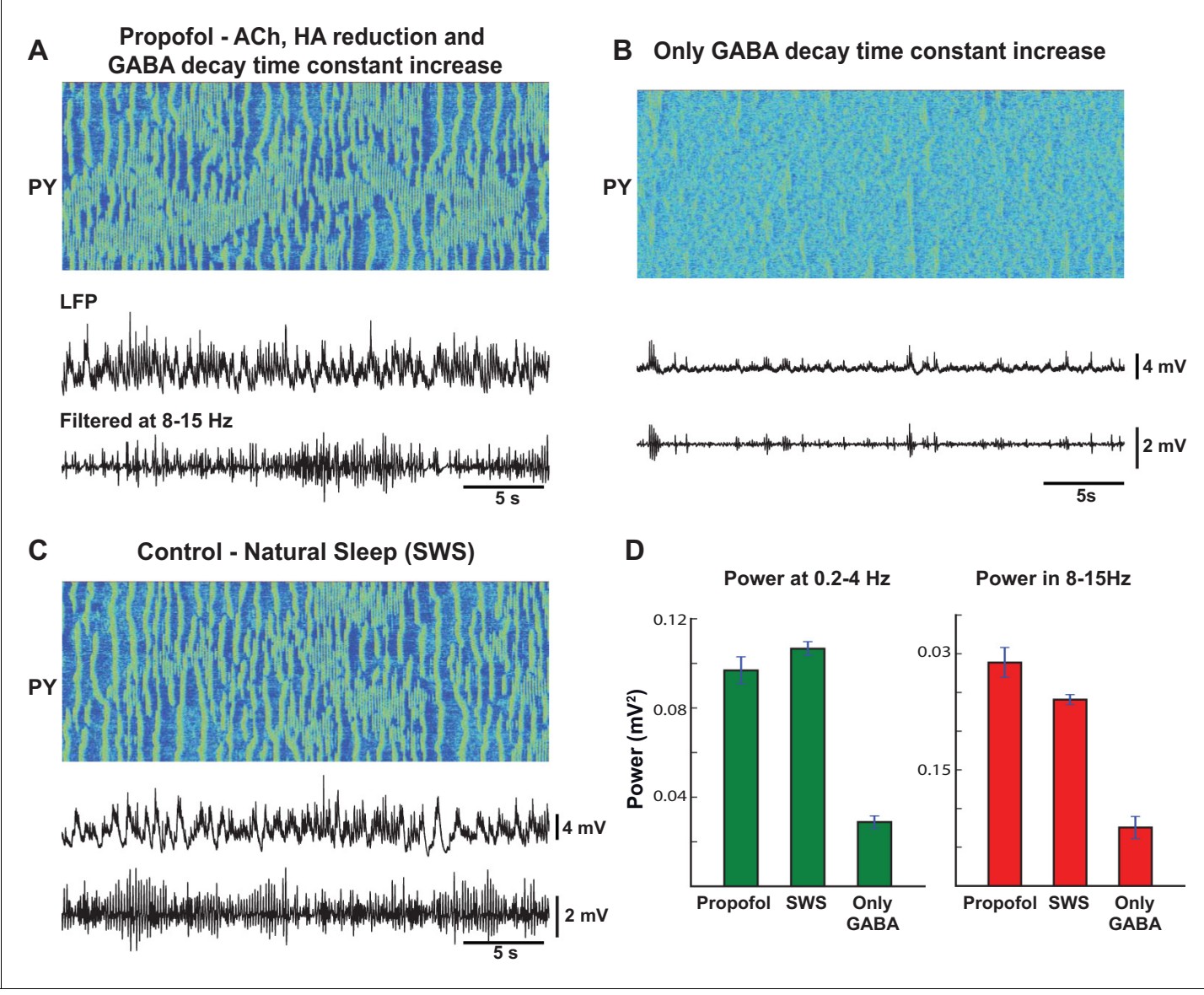

**Figure 9.** Mechanism of action of propofol induced anesthesia. (**A**) Model of propofol anesthesia implementing inhibition of ACh and HA release and increase in the decay time constant of the GABAergic IPSPs. From top to bottom: Spatio-temporal pattern of activity in PY neurons, average activity in PY network (simulated LFP), band-filtered (7–15 Hz) PY LFP. The GABA decay time constant was increased by 150%; ACh and HA were reduced to the same level as in simulations of the natural sleep in cats. Note significant amount of spindle activity. (**B**) The model where only the GABA time constant was increased by 150%. Note an almost complete absence of spindle activity. (**C**) Network activity during simulated natural SWS in cat model. Note decreased spindle activity compared to the propofol simulations in panel A. (**D**) Power in 0.2–4 Hz and 8–15 Hz bands for all three conditions. The network was divided into 10 groups of 50 neurons. Membrane voltages were averaged within each group, then FFT was used to estimate power in each group. Bar height indicates average across 10 groups, error bar indicates standard deviation across 10 groups.

oscillations in the thalamus of decorticated cats (*Timofeev and Steriade, 1996*) point to the primarily intracortical origin for this rhythm. (Note that thalamus can be actively involved in synchronization of the SO [*Lemieux et al., 2014*]). In vivo and in vitro studies suggest that the minimal substrate accounting for spindle oscillations (7–15 Hz recurrent oscillatory activity found during stage 2 (N2) of NREM sleep) is the interaction between thalamic reticular and relay cells (*Steriade and Deschénes, 1984*; *Steriade et al., 1985*; *Steriade and Llinas, 1988*, *1990*; *von Krosigk et al., 1993*). Previous computational studies proposed a minimal set of the mechanisms sufficient to model sleep spindles and slow oscillations (*Destexhe et al., 1996*; *Bazhenov et al., 1998*, *1999*; *Timofeev et al., 2000*;

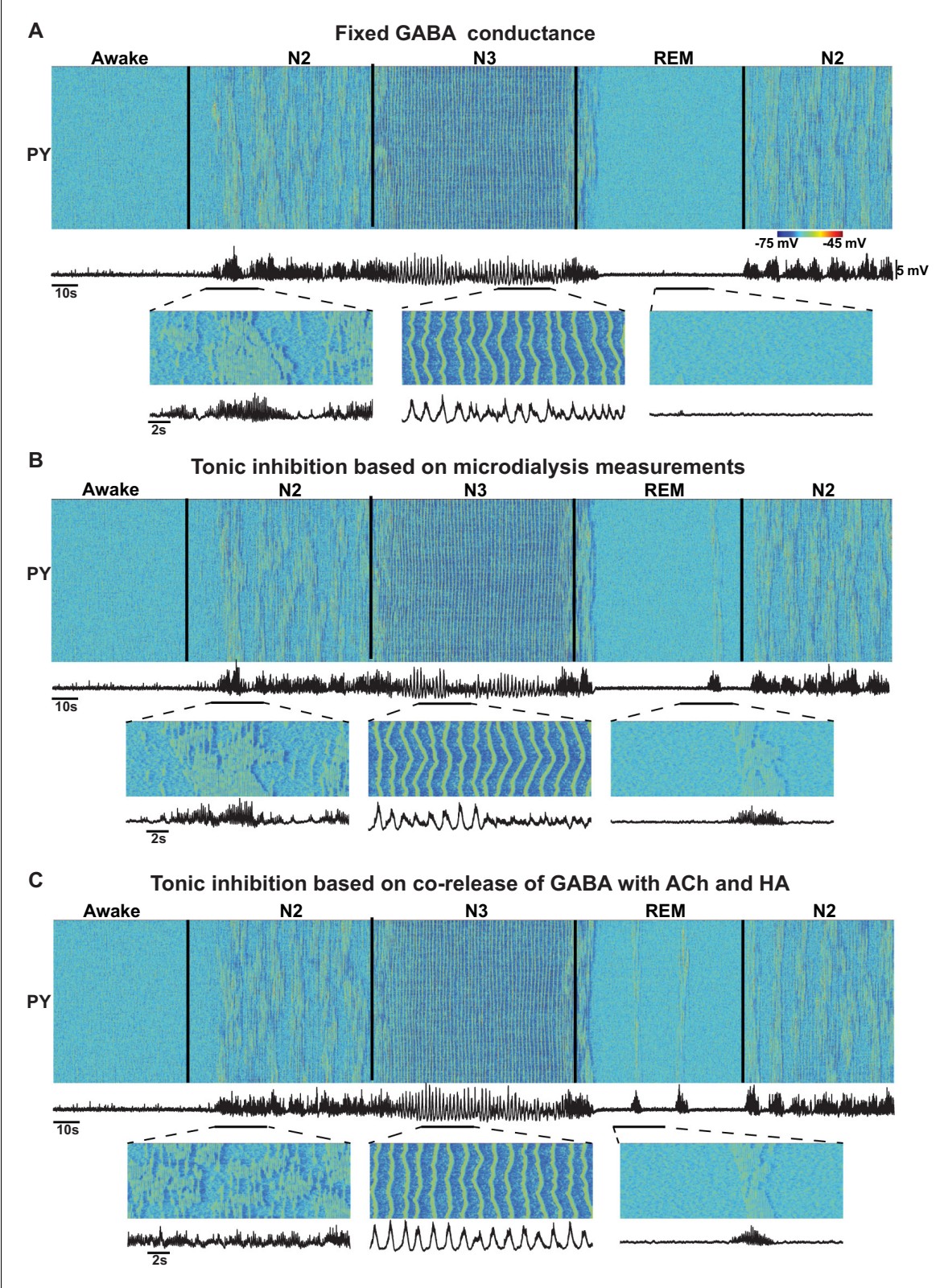

**Figure 10.** Effect of GABA on the network state transitions. (A) Both phasic and tonic GABA conductances were fixed for entire simulation. Note the state transitions similar to the baseline model (*Figure 2*). (B) Tonic GABA conductance (conductance of the miniature IPSPs) was varied based on the measured GABA levels in microdialysis experiments (mIPSPs were modified as 115%, 130% and 75% of awake stage for N2, N3 and REM stages, respectively), while phasic GABA conductance was remained fixed. (C) Assuming co-release of ACh with GABA, tonic GABA conductance was

*Figure 10 continued on next page*

*Figure 10 continued*

proportionally varied with ACh concentration (mIPSP for GABA conductance was scaled as 85%, 70% and 125% of awake stage for N2, N3 and REM stages respectively). Transition between sleep stages was observed in all these conditions, but spindles were less synchronous in C; in A and B alpha-bursts were less common.

*Bazhenov et al., 2002*; *Compte et al., 2003*; *Hill and Tononi, 2005*; *Bonjean et al., 2011*). An increase in the $K^+$ leak current was identified as a critical component for the transition between awake and slow wave sleep (*Bazhenov et al., 2002*; *Hill and Tononi, 2005*), and it was predicted that synchronization of the slow oscillation depends on cortico-cortical connections (*Hill and Tononi, 2005*). Several phenomenological and reduced mathematical models have been proposed to test the effects of neuromodulators on the sleep-wake cycle (*McCarley and Hobson, 1975*; *Robinson et al., 2011*; *Schellenberger Costa et al., 2016*). Nevertheless, none of these previous models examined transitions among all major sleep stages based on biophysical mechanisms.

In this study, we found that reduction in HA and ACh in the thalamus was required for induction of N2-like activity, characterized by recurrent spindles with characteristics consistent with animal and human data. Other neuromodulatory changes (such as change of GABA conductance) were not sufficient alone to induce N2-like activity, but played a role in determining the power and the synchrony of oscillations. Reduction in the ACh levels within the cortical network was identified to be a minimal requirement for the transition from the N2 (spindles) to the N3 (slow oscillation). Our study of the minimal models revealed the importance of the cell-type-specific action of the neuromodulators. Indeed, in order to model sleep stage transitions, the crucial changes were in HA, primarily acting on thalamic neurons, and in ACh, primarily acting within a cortex. All the network rhythms reported in this study resulted from intrinsic (autonomous) dynamics of the thalamocortical network; no external stimulation was applied to entrain any of the oscillatory activities in the model.

By parametrically varying the levels of the ACh, HA and GABA, we observed a wider range of activities in the thalamocortical network than have been previously reported in the computational models, but which do correspond to those that have been observed experimentally. While we found network states dominated by either spindles or slow oscillations, under certain conditions (such as only moderate reduction of the ACh level) we also observed mixed states combining these two major rhythms. Such mixed states may occur in vivo under normal conditions (*Aeschbach and Borbely, 1993*; *Muller et al., 2006*); our recent behavior study suggests that the strengths of the phase amplitude coupling between the spindle and the slow oscillation during NREM sleep correlates with memory consolidation (*Niknazar et al., 2015*). We speculate that the mixed states may become prevalent in some pathological conditions such as in Alzheimer's disease where sleep is altered and changes to the neuromodulatory system are reported (*Wulff et al., 2010*). Furthermore, our results on the putative relationship between levels of neuromodulators and characteristic sleep electrical activities could be potentially relevant to understanding the changes observed with aging. Indeed, the level of the HA in cerebrospinal fluid is known to be elevated with aging (*Prell et al., 1988*), where spindle power is known to be reduced (*Crowley et al., 2002*). In our model, the HA level is one of the major predictors of spindle power. Thus, we can speculate that an increase of HA with age could contribute to the decline of the spindle power.

In our study, the effects of neuromodulators were limited by changing the following four model properties – maximal conductances of the $K^+$-leak, $I_h$, GABA and AMPA currents. We found that specific combinations of changes of these core parameters were sufficient to cause transitions between sleep stages. We also tested a more detailed model, implementing the effects of ACh and HA on the other $K^+$ currents as well as on the persistent $Na^+$ and potassium M-currents (*Constanti and Galvan, 1983*; *McCormick and Williamson, 1989*; *Mittmann and Alzheimer, 1998*). We observed that this extended model qualitatively captured results similar to the minimal model. In contrast, after fixing the $K^+$-leak, $I_h$, GABA and AMPA currents but leaving the action of neuromodulators on the other channels, the model failed to show transitions between sleep stages, suggesting that the cellular and synaptic properties identified in our study are critical in determining patterns of electrical activity in the thalamocortical network.

In developing our model, we assumed that the levels of ACh, GABA and HA were different between N2 (Stage 2) and N3 (Stage 3 or SWS). These assumptions are consistent with data on

spiking activity in the Basal Forebrain region (*Aston-Jones and Bloom, 1981; Szymusiak et al., 2000*), suggesting a difference in ACh between the N2 and N3. Another assumption used in this study was that ACh modulates the strength of the excitatory AMPA synapses in the intracortical and thalamocortical circuits. Indeed, ACh suppresses the spread of excitation in vivo (*Kimura et al., 1999*) and reduces evoked responses (*Chauvette et al., 2012*). In slice experiments, ACh activation of muscarinic receptors suppressed both intracortical and thalamocortical excitatory connections, while nicotinic activation led to suppression of the intracortical connections and enhancement of the thalamocortical connections (*Gil et al., 1997*; *Hsieh et al., 2000*).

By comparing recordings from animals (cats and mice LFP) and human ECoG data across sleep stages, we found a significant difference in the interaction between major sleep rhythms (spindles and slow oscillation) during NREM sleep. In cats and mice, there was a strong positive correlation between spindle and delta power, while in humans there was a negative correlation between oscillations in these frequency bands. This difference may reflect the prominence of N2 (stage 2) in humans, in whom N2 has the longest duration of any sleep stage. In contrast, in cats, N2 is very rare compared to N3. In humans, N2 is mainly characterized by spindles with relatively short periods of slow oscillation or isolated K-complexes. (K-Complexes are large downward deflections in EEG often followed by spindle activity and is one of the electrophysiological markers of N2 (*Loomis et al., 1937*; *Amzica and Steriade, 1997*; *Cash et al., 2009*). In N3, spindles continue to occur in conjunction with the slow oscillation, but the EEG is dominated by slow activity. Our model predicts that this inter-species difference can be explained by a difference in the relative level of ACh during NREM sleep. This result is consistent with microdialysis measurements in cats where ACh was present during SWS albeit at low concentrations compared to wakefulness (*Marrosu et al., 1995*).

Surprisingly, we found that even minor changes of neuromodulators may lead to significant changes in the characteristics of brain electrical activity during sleep. Our model predicts that fluctuations of the level of ACh over time may explain the appearance of short periods of slow oscillations in N2 in humans (*Achermann and Borbely, 1997*) or modulate gamma activity during Up-states of the slow oscillation (*Mena-Segovia et al., 2008*). We hypothesize that transient activities such as K-Complexes during N2 sleep or PGO waves during REM stage may arise from fluctuations in ACh and/or other neuromodulators. Ultimately, our study predicts that transient changes in the level of neuromodulators may affect the interaction and characteristic properties of sleep rhythms and may thus affect functional outcomes of sleep, such as the impact of sleep on memory and learning.

## Materials and methods

### Model description

#### Neuromodulators and sleep stages

Our model implements a set of intrinsic and synaptic mechanisms that are known to be related to effects of ACh, HA and GABA. Specifically, ACh modulated the potassium leak currents in all neurons and the strength of excitatory AMPA connections in the cortex; HA modulated the strength of the hyperpolarization-activated cation current, $I_h$, in thalamic relay neurons; and GABA modulated the maximal conductance of the GABAergic synapses in cortex and thalamus. Based on experimental data, when compared to the awake stage, the levels of ACh and HA were reduced during NREM sleep (N2 and N3) and increased during REM sleep (*Vanini et al., 2012*). Concentration of the extracellular GABA in the neocortex is higher during NREM sleep and reduced during REM sleep compared to the awake state (*Vanini et al., 2012*). The origin of this change in the level of GABA is not fully understood. We examined the effect of changes in both phasic and tonic inhibitions. Thus, we tested three different models: (1) no change in the level of the GABA release; (2) miniature IPSPs were changed based on the measurements of extracellular GABA levels in microdialysis experiments (*Vanini et al., 2012*); and (3) miniature IPSPs were modified based on the assumption that GABA is co-released with ACh and HA, as shown in some experiments (*Saunders et al., 2015*; *Yu et al., 2015*).

## Intrinsic currents

The cortical neuron model included dendritic and axo-somatic compartments, similar to the previous studies (*Timofeev et al., 2000*; *Bazhenov et al., 2002*; *Chen et al., 2012*) and was described by the following equations,

$$C_m \frac{dV_d}{dt} = -\left[ACh_{PY} \cdot I_d^{K-leak}\right] - I_d^{leak} - I_d^{Na} - I_d^{Nap} - I_d^{Ca} - I_d^{KCa} - I_d^{Km} - I^{syn}$$
$$g_c^s(V_d - V_s) = -I_s^{Na} - I_s^{K} - I_s^{Nap}$$

where the subscripts $s$ and $d$ correspond to axo-somatic and dendritic compartments, $ACh_{PY}$ is the variable which is inversely proportional to the level of ACh in the cortex ($Ach_{PY} = 1$ for awake state, $ACh_{PY} = 1.25$ for N2, $ACh_{PY} = 1.8$ for N3 and $ACh_{PY} = 0.85$ during REM) and determines the strength of the potassium leak current ($I_d^{K-leak}$), $I_d^{leak}$ is the $Cl^-$ leak currents, $I^{Na}$ is fast $Na^+$ channels, $I^{Nap}$ is persistent sodium current, $I^K$ is fast delayed rectifier $K^+$ current, $I^{Km}$ is slow voltage-dependent non-inactivating $K^+$ current, $I^{KCa}$ is slow $Ca^{2+}$ dependent $K^+$ current, $I^{Ca}$ is high-threshold $Ca^{2+}$ current, $I^h$ is hyperpolarization-activated depolarizing current and $I^{syn}$ is the sum of the synaptic currents to the neuron. All intrinsic currents were of the form $g(V - E)$, where $g$ is the conductance, $V$ is the voltage of the corresponding compartment and $E$ is the reversal potential. The detailed descriptions of individual currents are provided in previous publications (*Bazhenov et al., 2002*; *Chen et al., 2012*). In the extended model (*Figure 11*), the conductances of $I_d^{KCa}$, $I_d^{Km}$ and $I_d^{Nap}$ were also scaled by $ACh_{PY}$ variable. The conductances of the leak currents were 0.007 mS/cm$^2$ for $I_d^{K-leak}$ and 0.023 mS/cm$^2$ for $I_d^{leak}$. The maximal conductances for the voltage and ion-gated intrinsic currents were, $I_d^{Nap}$: 2.0 mS/cm$^2$, $I_d^{Na}$: 0.8 mS/cm$^2$, $I_d^{Ca}$: 0.012 mS/cm$^2$, $I_d^{KCa}$: 0.015 mS/cm$^2$, $I_d^{Km}$: 0.012 mS/cm$^2$, $I_s^{Na}$: 3000 mS/cm$^2$, $I_s^{K}$: 200 mS/cm$^2$ and $I_s^{Nap}$: 15 mS/cm$^2$. $C_m$ was 0.075 µF/cm$^2$.

The following equation describes the cortical inhibitory (IN) neuron:

$$C_m \frac{dV_d}{dt} = -\left[ACh_{PY} \cdot I_d^{K-leak}\right] - I_d^{leak} - I_d^{Na} - I_d^{Ca} - I_d^{KCa} - I_d^{Km} - I^{syn}$$
$$g_c^s(V_d - V_s) = -I_s^{Na} - I_s^{K}$$

The conductances for the leak currents for IN neurons were 0.034 mS/cm$^2$ for $I_d^{K-leak}$ and 0.006 for $I_d^{leak}$. Maximal conductances for the active currents were, $I_d^{Na}$: 0.8 mS/cm$^2$, $I_d^{Ca}$: 0.012 mS/cm$^2$, $I_d^{KCa}$: 0.015 mS/cm$^2$, $I_d^{Km}$: 0.012 mS/cm$^2$, $I_s^{Na}$: 2500 mS/cm$^2$ and $I_s^{K}$: 200 mS/cm$^2$.

The thalamic relay (TC) neuron consisted of a single compartment,

$$\frac{dV}{dt} = -\left[ACh_{TC} \cdot I^{K-leak}\right] - I^{leak} - I^{Na} - I^{K} - I^{LCa} - I^{h} - I^{syn}$$

where $ACh_{TC}$ was 1, 1.25, and 0.85 during awake, N2, N3 and REM sleep correspondingly. The conductances of the leak currents were, $I^{leak}$: 0.01 mS/cm$^2$, $I^{K-leak}$: 0.007 mS/cm$^2$. The maximal conductance for the other currents were, fast $Na^+$ ($I^{Na}$) current: 90 mS/cm$^2$, fast $K^+$ ($I^K$) current: 10 mS/cm$^2$, low threshold $Ca^{2+}$($I^{LCa}$) current: 2.5 mS/cm$^2$, hyperpolarization-activated cation current ($I^h$): 0.015 mS/cm$^2$. The effect of HA on $I^h$ was implemented as a shift in the activation curve based on experimental evidences as follows,

$$I^h = g([O] + k[O_L])(V - E_h)$$

$$C \xrightarrow{\alpha(V)} O \qquad C \xleftarrow{\beta(V)} O$$

$$P_0 + 4Ca^{2+} \xrightarrow{k_1} P_1 \qquad P_0 + 4Ca^{2+} \xleftarrow{k_2} P_1$$

$$O + P_1 \xrightarrow{k_3} O_L \qquad O + P_1 \xleftarrow{k_4} O_L$$

$$\alpha(V) = h_\infty(V)/\tau_s(V)$$

$$\beta(V) = (1 - h_\infty(V))/\tau_s(V)$$

$$h_\infty(V) = \frac{1}{1 + e^{(V+75+Shift_{HA})/5.5}}$$

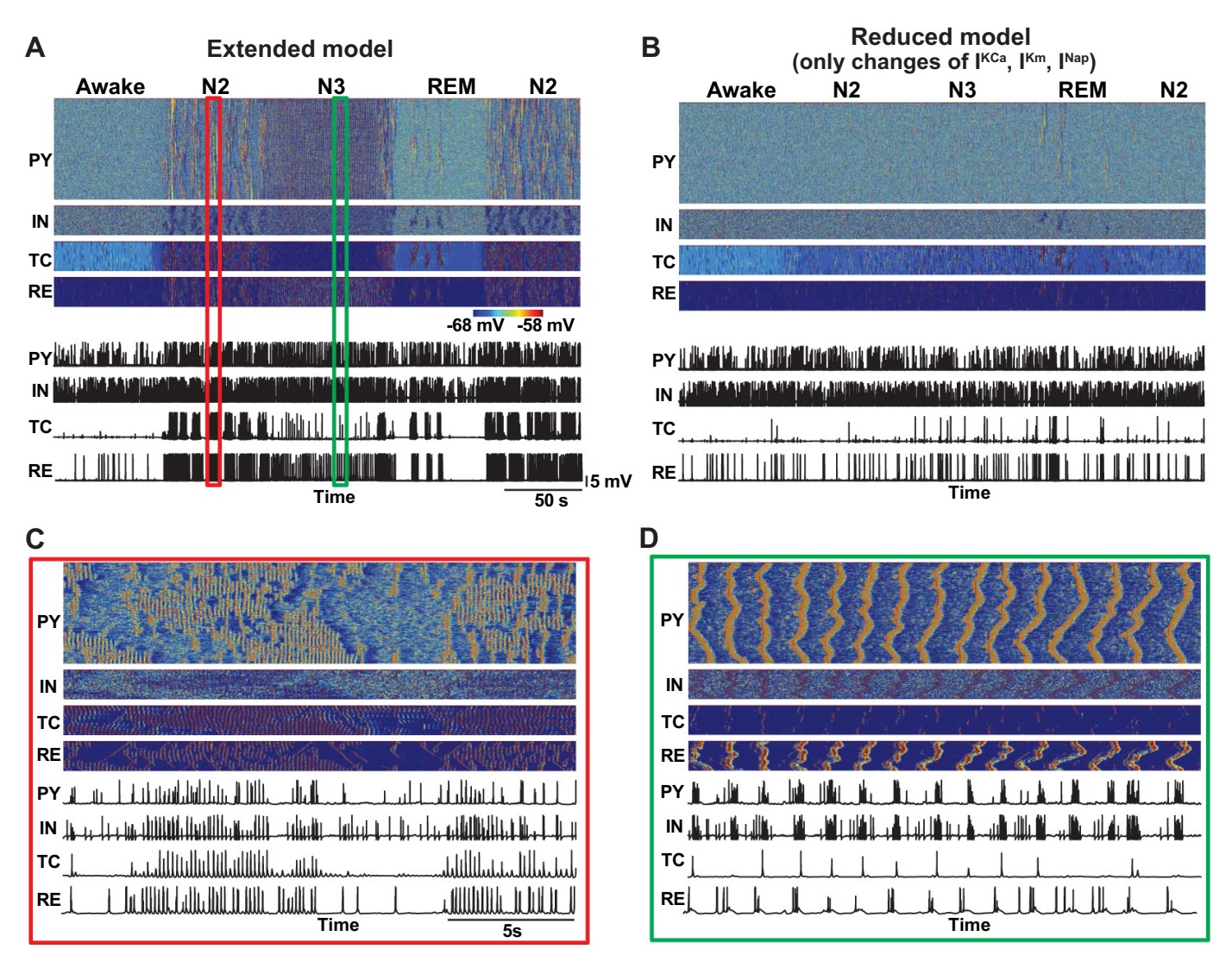

**Figure 11.** Effect of additional parameter changes on the stage transitions. (A) Network activity in an extended model with ACh and HA having effects on the all K$^+$ channels (Ca$^{2+}$-dependent K$^+$ current, M-current, K$^+$ leak current), persistent Na$^+$ current and H-current. (B) Network activity in the model implementing only additional effects of ACh and HA (Ca$^{2+}$-dependent K$^+$ current, M-current, persistent Na$^+$ current) but having core intrinsic and synaptic properties (K$^+$-leak, I$_h$, GABA and AMPA) fixed as in the awake model. This model failed to simulate transitions between sleep stages. (C and D) Zoom in of the periods during spindle (C) and slow oscillation (D) from panel A.

where $\mathrm{Shift}_{HA}$ was −8 mV, −3 mV, −2 mV and 0 mV for awake, stage 2, SWS and REM sleep correspondingly. The values of other constants were the same as in previous study (**Bonjean et al., 2011**).

The RE neuron was modeled as a single compartment as follows,

$$\frac{dV}{dt} = -\left[ACh_{RE} \cdot I^{K-leak}\right] - I^{leak} - I^{Na} - I^{K} - I^{Ca} - I^{syn}$$

where $ACh_{RE}$ was 1, 0.8, 0.5 and 1.15 for awake, N2, N3 and REM stage. Since it is known that ACh reduces excitability of the RE cells (**McCormick, 1992**), the ACh was reduced during NREM stage compared to awake. The conductances for leak current were, I$^{leak}$: 0.05 mS/cm$^2$, I$^{K-leak}$: 0.016 mS/cm$^2$. The maximal conductance for the other currents were, fast Na$^+$ (I$^{Na}$) current: 100 mS/cm$^2$, fast K$^+$ (I$^{K}$) current: 10 mS/cm$^2$, low threshold Ca$^{2+}$(I$^{LCa}$) current: 2.2 mS/cm$^2$.

## Network structure

Network architecture of the thalamocortical model was based on the previous studies of the spindle and slow-wave activity (*Bazhenov et al., 2002*; *Bonjean et al., 2011*; *Chen et al., 2012*; *Wei et al., 2016*). The thalamocortical network consisted of five hundred cortical pyramidal (PY) neurons, one hundred cortical inhibitory (IN) neurons, one hundred thalamic relay (TC) neurons and one hundred thalamic reticular (RE) neurons. The network connectivity between different groups of neuron is shown in *Figure 1A*. One-dimensional network topology was used in each layer and connectivity was local and determined by the radius of connections. The PY neurons received AMPA and NMDA synapses from other PY neurons within radius of 5. The TC neurons projected to RE neurons through AMPA synapses (radius of 8), and connections from RE to TC neurons included GABA-A and GABA-B synapses (radius of 8). Thalamocortical connections were wider and mediated by AMPA synapses (TC to PY: radius of 10; TC to IN: radius of 2); corticothalamic connections included AMPA synapses (PY to TC: radius of 10; PY to RE: radius of 8) and were stronger to reticular neurons than to thalamic relay neurons.

## Synaptic currents

GABA-A, NMDA and AMPA synaptic currents were described by first-order activation schemes (*Destexhe et al., 1994*). The equations for all synaptic currents used in this model are given in our previous publication (*Bazhenov et al., 2002*; *Chen et al., 2012*). Briefly, below we mention only the relevant equations that were used to model effect of the varying levels of ACh and GABA.

$$I_{syn}^{AMPA} = L_{Ach} \cdot g_{syn}[O](V - E_{AMPA})$$

$$I_{syn}^{GABA} = L_{GABA} \cdot g_{syn}[O](V - E_{GABA})$$

$L_{ACh}$ was 1, 1.25, 2.0 and 0.8 for awake, N2, N3 and REM stages respectively for cortical PY-PY, TC-PY and TC-IN connections; $L_{GABA}$ was 1, 1.15, 1.3 and 0.75 for awake, N2, N3 and REM stages respectively for cortical IN-PY, RE-RE and RE-TC connections. In these equations, $g_{syn}$ is the maximal synaptic conductance and $[O(t)]$ is the fraction of open channels, $E_{AMPA}(= 0$ mV$)$ and $E_{GABA}$ $(= -90$ mV$)$ are reversal potential for AMPA and GABA. The maximal conductances were (subscript indicates cell types and superscript indicates type of connection): $g_{PY-PY}^{AMPA}$ = 0.024 μS, $g_{PY-PY}^{NMDA}$ = 0.001 μS, $g_{PY-TC}^{AMPA}$ = 0.005 μS, $g_{PY-RE}^{AMPA}$ = 0.015 μS,, $g_{PY-IN}^{AMPA}$ = 0.012 μS, $g_{PY-IN}^{NMDA}$ = 0.001 μS, $g_{IN-PY}^{GABA-A}$ = 0.024 μS, $g_{TC-CX}^{GABA-A}$ = 0.02 μS, $g_{TC-IN}^{GABA-A}$ = 0.02 μS, $g_{RE-RE}^{GABA-A}$ = 0.05 μS, $g_{RE-TC}^{GABA-A}$ = 0.05 μS, $g_{RE-TC}^{GABA-B}$ = 0.02 μS and $g_{TC-RE}^{AMPA}$ = 0.05 μS. Detailed description of the O(t) is given in a previous publication (*Bazhenov et al., 1998*; *Timofeev et al., 2000*) and is based on first-order activation schemes. In addition, spontaneous miniature EPSPs and IPSPs were implemented for PY-PY, PY-IN and IN-PY connections. The synaptic dynamics followed the same equations as the regular PSPs, and their arrival times were modeled by Poisson processes with time-dependent mean rate, with next release time given by function: $(2/(1 + \exp(-(t-t_0)/20))-1)/250$, where $t_0$ is a time instant of the last presynaptic spike. The maximal conductance for miniature PSP were: $g_{mini(PY-PY)}^{AMPA}$ = 0.033 μS, $g_{mini(PY-IN)}^{AMPA}$ = 0.02 μS and $g_{mini(IN-PY)}^{AMPA}$ = 0.02 μS. Short-term depression of intracortical excitatory connections was also included, similar to previous publications (*Bazhenov et al., 2002*; *Chen et al., 2012*).

## Stimulation protocol

A brief DC input lasting for 100 ms and amplitude of 1.5 nA was applied to TC neurons during awake state in some simulations to test stability of the network dynamics. The DC pulse was modeled using two Heaviside functions. No periodic stimulation was applied in any of the modeling experiments.

## In vivo experiments in cats

Experiments were conducted on adult non-anesthetized cats. The cats were purchased from an established animal breeding supplier. Good health conditions of all animals were certified by the supplier and determined upon arrival to the animal house by physical examination, which was performed by animal facilities technicians and a veterinarian in accordance with requirements of the

Canadian Council on Animal Care. The protocol for experiments involving cats was approved by CPAUL, Comité de protection des animaux de l'Université Laval (Protocol # 2012–174-4). The surgery was performed on animals 5–20 days after their arrival to the local animal house. We recorded field potentials from several cortical areas and from the VPL thalamic nucleus of cats during natural sleep/wake transitions.

## Preparation for cat surgeries

Chronic experiments were conducted using an approach similar to that previously described (*Steriade et al., 2001*; *Timofeev et al., 2001*). For implantation of recording chamber and electrodes, cats were anesthetized with isoflurane (0.75–2%). Prior to surgery, the animal was given a dose of preanesthetic, which was composed of ketamine (15 mg/kg), buprenorphine (0.01 mg/kg) and acepromazine (0.3 mg/kg). After site shaving and cat intubation for gaseous anesthesia, the site of incision was washed with at least three alternating passages of a 4% chlorexidine solution and 70% alcohol. Lidocaine (0.5%) and/or marcaine (0.5%) was injected at the site of incision and at all pressure points. During surgery, electrodes for LFP recordings, EMG from neck muscle, and EOG were implanted and fixed with acrylic dental cement. Eight to ten screws were fixed to the cranium. To allow future head-restrained recordings without any pressure point, we covered four bolts in the dental cement that also covered bone-fixed screws and permanently implanted electrodes. Throughout the surgery, the body temperature was maintained at 37°C using a water-circulating thermoregulated blanket. Heart beat and oxygen saturation were continuously monitored using a pulse oximeter (Rad-8, MatVet) and the level of anesthesia was adjusted to maintain a heart beat at 110–120 per minute. A lactate ringer solution (10 ml/kg/hr, intravenously [i.v.]) was given during the surgery. After the surgery, cats were given buprenorphine (0.01 mg/ kg) or anafen (2 mg/kg) twice a day for 3 days and baytril (5 mg/kg) once a day for 7 days. About a week was allowed for animals to recover from the surgery before the first recording session occurred. Usually, 2–3 days of training were sufficient for cats to remain in head-restrained position for 2–4 hr and display several periods of quiet wakefulness, SWS, and REM sleep. The recordings were performed up to 40 days after the surgery.

## In vivo recordings in cats

All in vivo recordings were done in a Faraday chamber. LFPs were recorded using tungsten electrodes (2 MΩ, band-pass filter 0.1 Hz to 10 kHz) and amplified with AM 3000 amplifiers (A-M systems) with custom modifications. We aimed to implant electrodes at 1 mm below the cortical surface. A silver wire was fixed either in the frontal bone over the sinus cavity or in the occipital bone over the cerebellum and was used as a reference electrode. All electrical signals were digitally sampled at 20 kHz on Powerlab (ADinstruments, Colorado Springs, USA) and stored for offline analysis. At the end of the experiments, the cats were euthanized with a lethal dose of pentobarbital (100 mg/kg, i.v.).

## In vivo experiments in mice

Experiments were conducted on adult C57Bl6 mice. The implantation of recording electrodes was done under isoflurane anesthetized (0.75%–2%). After site shaving, the site of incision was washed with at least three alternating passages of a 4% chlorexidine solution and 70% alcohol. Lidocaine (0.5%) and/or marcaine (0.25%) was injected at the site of incision and at all pressure points. Prior to surgery, the animal was given a dose of buprenorphine (0.05–0.1 mg/kg). During surgery, electrodes for LFP recordings (custom-made, stainless steel) and EMG from neck muscle were implanted and fixed with acrylic dental cement. We aimed at implanting LFP electrodes at a depth of about 650 μm. Four or five anchoring screws were fixed to the cranium. A stainless steel screw implanted above the cerebellum was used as a reference. Throughout the surgery, the body temperature was maintained at 37°C using a water-circulating thermoregulated blanket. Saline (0.1–0.5 ml) and anafen (5 mg/kg) subcutaneous injection were performed at the end of the surgery. After the surgery, mice were given anafen (5 mg/kg) once a day for 2 days. About a week was allowed for animals to recover from the surgery before the first recording session occurred. LFPs were band-pass filtered (0.1–100 Hz). Freely moving mice were plugged to a custom-made headstage to reduce movement artifacts and then amplified with AM 3000 amplifiers (A-M systems). All electrical signals were digitally sampled at 1 kHz on Powerlab (ADinstruments, Colorado Springs, USA) and stored for offline

analysis. At the end of the experiments, mice were euthanized with a lethal dose of euthanyl (120 mg/kg, i.v.). The protocol for mice experiments was approved by CPAUL, Comité de protection des animaux de l'Université Laval (Protocol # 2013-014-4).

### Data analysis for animal recordings

Electrographic recordings in cats and mice (shown in *Figure 6*) were analyzed offline using custom-written routines in the IgorPro software. The delta power was calculated from 5 s time windows, using 1 s sliding time window, as the integral power between 0.2 and 4 Hz of the full spectrogram; the spindles power was calculated as the integral power between 8 and 15 Hz. We also computed spindle and delta power using longer time windows (2 s, 5 s, and 30 s). For scatter plots and correlation analysis in *Figure 6*, spindle and delta power was computed from 30 s windows in 0.2–4 Hz and 8–15 Hz frequency bands. In *Figure 7B* a shorter time window of 2 s was used. In all cases, we found significant positive correlation between spindle and delta power as reported in the Results section.

### ECoG recordings and analysis

Recordings were obtained during natural sleep in a patient undergoing monitoring with intracranial electrodes in order to locate spontaneous seizure onset and guide surgical treatment. Recordings were obtained after informed consent monitored by the Partners Human Subject Protection Committee (Protocol #2007P00165). The patient had an $8 \times 8$ 64-contact peri-sylvian grid and four strips implanted subdurally over the left hemisphere, sampling temporal, Rolandic, parietal and frontal cortices at 1 cm spacing. After rejection of electrodes that either had poor contact or showed epileptic interictal activity, such as interictal spikes, 42 electrodes remained for analysis. We analyzed a recording period of 9 hr at night and sleep scored the patient based on 30 s epochs. Staging was scored using information from scalp, EOG and intracranial electrodes and standard criteria. Awake, REM, N2, and N3 periods were used for further analysis. A fast Fourier transform with a window size of 30 s was performed and power in the SO (0.01–2 Hz) and spindle (9–17 Hz) band frequencies was obtained, and their correlations calculated for 30 s epochs during N2 and N3 periods The average Pearson's correlation coefficient R value was −0.2592 and standard deviation of 0.2281. The average R-squared was 0.1181. 15 electrodes showed significant negative correlation compared to 0 electrodes showing positive correlation after Bonferroni Correction at the p=0.05 level (n = 42, 0.05/42 = 0.0012 = corrected p-value for significance). When focusing on electrodes over the postcentral gyrus (GR7, GR8, GR12, GR14, GR15, GR16, GR21, GR22, GR23, GR24), the average R was -0.4657 with a standard deviation of 0.1721.

### Cluster analysis

To compare results of network simulations based on different combinations of model parameters, we performed cluster analysis on the phase locking value (in frequency range 0.5–20Hz) for spindle (7–15Hz) and delta (0.5–4Hz) power. Each trial was 20 s duration; however, the first 10 s was discarded to avoid transient activity. We used Akaike information criteria (AIC) (*Akaike, 1974*) to measure the least number of components that are required to fit the data. This analysis resulted in a minimum of 10 components beyond which the AIC information measure did not increase. Next a Gaussian mixture model fitting with 10 components using EM method with 500 iterations was applied to obtain the clusters (corresponds to different colored points in *Figure 4A*). We then determined the projection of the clusters to the space of neuromodulators and finally fitted an ellipsoid to provide an approximate location of the clusters in the neuromodulator space. The ellipsoid fit was performed using geom3d library (https://github.com/dlegland/matGeom/wiki/geom3d). The power, cluster analyses and 3D plotting were performed using MATLAB (MathWorks Inc.). Simulations for various values of neuromodulators were supported by computational resources from the neuroscience gateway (*Sivagnanam et al., 2013*).

## Acknowledgements

This work was supported by grants from ONR (MURI: N000141310672) and NIH (MH099645 and EB009282), Canadian Institutes of Health Research (grants MOP-136969, MOP-136967) and by National Sciences and Engineering Research Council of Canada (grant 298475).

## Additional information

### Funding

| Funder | Grant reference number | Author |
|---|---|---|
| Canadian Institutes of Health Research | MOP-136969 | Igor Timofeev |
| Canadian Institutes of Health Research | MOP-136967 | Igor Timofeev |
| Office of Naval Research | MURI: N000141310672 | Sydney S Cash<br>Eric Halgren<br>Maxim Bazhenov |
| National Institutes of Health | MH099645 | Eric Halgren<br>Maxim Bazhenov |
| National Institutes of Health | EB009282 | Eric Halgren<br>Maxim Bazhenov |
| National Sciences and Engineering Research Council of Canada | 298475 | Igor Timofeev |

The funders had no role in study design, data collection and interpretation, or the decision to submit the work for publication.

### Author contributions

GPK, MB, Conception and design, Analysis and interpretation of data, Drafting or revising the article; SC, IS, SS, IT, SSC, EH, Acquisition of data, Analysis and interpretation of data, Drafting or revising the article

### Author ORCIDs

Giri P Krishnan, http://orcid.org/0000-0002-3931-7633

### Ethics

Human subjects: The data were collected part of other study where informed consents were obtained. This is primarily a modeling paper and the human sleep data are used to illustrate a point. Animal experimentation: The data were collected part of other study where ethical guidelines were followed. This is primarily a modeling paper and the cat and mice sleep data are used to illustrate a point.

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
