## [Decision Letter]

Thank you for submitting your article "Cellular and Neurochemical Basis of Sleep Stages in the Thalamocortical network" for consideration by *eLife*. Your article has been reviewed by three peer reviewers, one of whom, Ronald L Calabrese (Reviewer #1), is a member of our Board of Reviewing Editors and the evaluation has been overseen by David Van Essen as the Senior Editor. The following individuals involved in review of your submission have agreed to reveal their identity: Paul Garcia (Reviewer #2); Miles Whittington (Reviewer #3).

The reviewers have discussed the reviews with one another and the Reviewing Editor has drafted this decision to help you prepare a revised submission.

Summary:

In this very interesting study that combines computational modeling and some in vivo recordings of brain activity during sleep and wakefulness and under ketamine anesthesia, the authors propose a comprehensive model of how major neuromodulators may control sleep states and their transitions in mammals. The model is grounded in previous experimental evidence and covers the major neuromodulators ACh, GABA, and HA. They show how these modulators might synchronize the spindle (N2) and slow oscillations, SO, (N3) states, as well as REM. Using ECoG recordings from humans and LFP recordings from cats and mice, during NREM sleep they show that the power of spindle and SO is negatively correlated in human and positively correlated in cats and mice, and explain this discrepancy by the differences in the relative levels of ACh. They also explore similarities and differences between SO during N3 and ketamine anesthesia and explain them in terms of influences of ketamine on neuromodulator levels and effects. The study identifies potential intrinsic and synaptic mechanisms through which neuromodulators acting in combination may mediate transitions between sleep stages. These findings should be of wide interest to the sleep, and anesthesia communities and to those interested in coordinated whole brain activity.

Essential revisions:

There are some major concerns about claims but overall this appears to be a strong study of general significance.

1) The authors have gone too far when suggesting that they are modelling the sleep-related effects of the neuromodulators they have chosen. The authors are really basing their conclusions on an oversimplified (and under-developed) link between ACh and effects on membrane currents. Their models do show remarkable state changes with manipulation of the 4 main parameters (K-leak, Ih, GABA, AMPA) and these finding are both interesting and challenging to the field – but the relation back to ACh and HA is not as strong as they imply. There are many precedents for ACh and HA having the opposite effect on the main model parameters on certain cell subtypes cf. the ones they use.

The authors should clarify that they are making a selection of potential modulator effects, and in Discussion suggest this MAY relate to known endogenous brain-state modulators like ACh, Histamine (and many others).

Alternatively the authors can revisit their model and represent more accurately the known effects of HA and ACh on each of the 'loose' cell types included in the model.

2) A similar concern applies to the handling of data gathered with ketamine anesthesia. The authors attempt to mimic ketamine in their model by decreasing NMDA's influence – but ketamine is a promiscuous molecule that has influence on a great number of channels. Additionally, the unconsciousness produced by the "dissociative" anesthetic, ketamine, has a very different phenotype than that of propofol or isoflurane which more or less produce a quiescence resembling sleep. As opposed to these GABAergic anesthetics, ketamine does not activate the sleep nuclei of the hypothalamus (VLPO), does not depress thalamic activity, and activates wake-promoting nuclei in the brain stem. The adjunct agent used in their in vivo experiments (xylazine) does on the other hand promote sleep, but does not do this through NMDA antagonism. Therefore the specificity of their model is really in question. George Mashour has done work on ketamine and his work might be better referenced. http://www.ncbi.nlm.nih.gov/pmc/articles/PMC4076669/

Further, we would expect that an increase in inhibition in the model would produce slow oscillations similar to sleep. This model state could be compared to in vivo data using propofol or isoflurane. Perhaps, this model is "tuned" to produce these stable oscillations in a variety of different parameter regimes – but as mentioned in the expert reviews (Reviewer #2), without an evaluation of the stimulation protocols it is difficult to determine what is a network effect vs. a reflection of the periodicity already present in the stimulation.

The expert reviews of reviewers #2 and #3 are appended to aid in the revision.

*Reviewer #2:*

This manuscript reports the results of a computational model of sleep cycling based on variations in neurotransmitters. The model was also compared to neuronal oscillations during sleep using human, cat and mouse datasets. The main conclusion is that the initiation of the oscillatory pattern indicative of N2 sleep can be simulated by a decrease in histamine and acetylcholine. A further decrease in acetylcholine can also explain a transition to deeper sleep. In general, this model contributes much to our understanding of the neurochemical basis of sleep macroarchitecture. Additionally, the results challenge theories that increases in the inhibitory neurotransmitter GABA mediate transitions among sleep stages. Although some of the conclusions need to be amended and there are several places that need some grammatical corrections, there is much to be admired in this paper. However some of the grammatical errors and non-sequitors (non-explored ideas) make me concerned that it was hastily prepared.

Summary:

– Some parts of the summary are awkward in regard to sentence structure and punctuation. The second sentence does not sound like proper subject verb agreement, and the third sentence requires the word and between the two subjects to make sense. Α/theta bursts are mentioned in the Summary but not emphasized in the paper (previous version?) please remove.

Introduction:

– First sentence of second paragraph should be reworded so characterize is not repeated. The first sentence of the third paragraph should start with: "Neuromodulators".

Results

– The TC stimulation is not represented in Figure 2. This input is also not described in the methods, given the strong excitatory connections between thalamus and cortex the nature of the stimulation ("simulating sensory input") could be driving the oscillations and must be included to evaluate the main conclusions of the paper.

– Why did they examine such a large frequency range to evaluate phase-locking? When the connections are not completely known (in vivo recordings) phase-locking among narrow frequency bands (especially among specific regions, e.g., thalamus and cortex) is much more important for synchronization. Very little of the quantitative results rely on phase-locking.

– Why is ketamine compared to SWS? Others report it as sharing similarities with REM. Propofol or isoflurane anesthesia would be a better comparison.

Methods

– Extrasynaptic and Intrasynaptic GABA concentrations are mentioned but not explored in the paper, please remove (unless added to discussion as to a potential limitation of the model).

– Network description: radius is not defined explicitly.

– The in vivo description mentions non-anesthetized cats and ketamine is given?

Discussion

–- Much of the Discussion can be considerably shortened.

– The Discussion does not adequately address other potential influences on spindle power (e.g., age and cortical volume).

*Reviewer #3:*

This is a very interesting paper that uses a fairly detailed computational model, constrained by experimental observations in a variety of species, which is used to demonstrate the dependence of sleeps stage on a variety of network and intrinsic cell properties (leak current, AMPA, GABA, I(h)). The paper extends the findings to link sleep stages to two main neuromodulators known to be involved in controlling sleep – Ach and Histamine.

In general the paper is well written and the data beautifully presented. I very much liked the demonstration of different sleep stages with different intrinsic properties but it would have been good to refer to the very detailed model of Hill & Tononi which, a while ago, came to very similar conclusions.

My only major problem with the manuscript comes from the attempt to relate the model parameter changes to histamine and Acetylcholine. I think a bit more detail and biological realism needs to be included here. For example, Ach is stated to affect AMPA-mediated transmission and leak current. Of the parameters used to construct the model cells it also affects Km and GABA release. Reading the methods it seems Ach was modelled as a scale factor for all intrinsic and synaptic properties – but this scale factor was increased for decreased Ach levels. This means, for example, that properties shown not to be affected by Ach were affected in the model, and properties of Ach well documented (such as depolarisation of main interneuron subtypes) were omitted or modelled conversely.

On a similar note, it is not clear how the 2 modulators chosen fit with the GABA levels used. There is a wealth of evidence linking Ach and Histamine to GABA release and postsynaptic effects – some which fit the basic hypothesis the authors propose (decreased Ach, increased GABA release), and some antagonistic to it (increased histamine, increased tonic GABA). This could be clarified with a more rigorous modelling of the known effects of the two agents to the conductances used in the model. If no changes in sleep stages are seen then this would provide evidence for an involvement of the other major neuromodulators involved in sleep not considered here.

[Editors' note: further revisions were requested prior to acceptance, as described below.]

Thank you for resubmitting your work entitled "Cellular and Neurochemical Basis of Sleep Stages in the Thalamocortical network" for further consideration at *eLife*. Your revised article has been favorably evaluated by David Van Essen (Senior editor), a Reviewing editor, and two reviewers.

The reviewers consider the manuscript to be much improved. There is only one minor issue that you may want to address before final acceptance, as outlined below:

Figure 1 Vertical axes might be clearer if they read "Normalized Conductance (%)" It should be obvious they are normalized to the Awake condition by the graph.

*Reviewer #2:*

This is a re-submission on a computational model of transitions among sleep stages that is firmly based in neurophysiology. The model is compared to neurophysiologic recordings from LFP records obtained in animal models and human ECoG data. Finally, the model's predictions are tested with simulations designed to mimic propofol anesthesia. This revised manuscript has much improved readability in both the precision of writing and clarity of thought. The main conclusions are that a reduction in acetylcholine and histamine levels in the thalamus corresponds to N2 sleep, characterized by spindle activity. And that an increase in GABA alone does not produce the oscillations characteristic of N2. The experiments with propofol anesthesia demonstrate this nicely. The model also predicts that reduced cortical acetylcholine influences the transition to N3 sleep. Overall, this is a nice contribution to the sleep and anesthesia literature.

*Reviewer #3:*

This one of the best, most comprehensive revisions of a paper I have seen. The authors have taken on every major comment and provided a great deal more clarity, focus and detail. I have no hesitation in recommending it for publication now.

---

## [Author Response]

[…]

*Essential revisions:*

*There are some major concerns about claims but overall this appears to be a strong study of general significance.*

1) The authors have gone too far when suggesting that they are modelling the sleep-related effects of the neuromodulators they have chosen. The authors are really basing their conclusions on an oversimplified (and under-developed) link between ACh and effects on membrane currents. Their models do show remarkable state changes with manipulation of the 4 main parameters (K-leak, Ih, GABA, AMPA) and these finding are both interesting and challenging to the field – but the relation back to ACh and HA is not as strong as they imply. There are many precedents for ACh and HA having the opposite effect on the main model parameters on certain cell subtypes cf. the ones they use.

*The authors should clarify that they are making a selection of potential modulator effects, and in Discussion suggest this MAY relate to known endogenous brain-state modulators like ACh, Histamine (and many others).*

As requested, we made a number of changes to the text to clearly indicate that we only tested a subset of the effects of neuromodulators and many known effects of neuromodulators were not included in our model. As pointed out by the reviewer, the four main parameters (K-leak, AMPA, Ih, GABA) were sufficient to explain transitions between sleep stages in the model. These cellular and synaptic properties were chosen based on the past experiments where they have been linked to the effects of neuromodulators. This putative (and certainly incomplete) link allowed us to interpret our results and to make specific predictions for future studies on the variations in the neuromodulators during different sleep stages and on the differences in the neuromodulatory response between humans and cats/mice. To further address reviewer concerns, we also examined transition between sleep stages in a more detailed model (see our response below), which revealed similar qualitative results as the reduced “four-parameter” model. The way we want to interpret this is that a simple model identifies the core or minimally required changes, which are likely to be linked to the known effects of the neuromodulators in the thalamocortical network. We include the relevant discussion in the modified manuscript, please see below.

The text in the result section was modified as follows:

“While our model does not capture the entire spectrum of changes associated with known effects of neuromodulators, we identified a minimal set sufficient to account for characteristic changes of brain electrical activity across the sleep-wake cycle. […] We also tested in the model the effect of extracellular GABA concentration on tonic inhibition.”

The following paragraph was included to the Discussion section:

“In our study, the effects of neuromodulators were limited by changing the following four model properties – maximal conductances of the K^+^-leak, Ih, GABA and AMPA currents. […] In contrast, after removing the changes of the K^+^-leak, Ih, GABA and AMPA currents but leaving the action of neuromodulators on the other channels, the detailed model failed to show transitions between sleep stages (Figure 11), suggesting that the cellular and synaptic properties identified in our study are critical in determining patterns of electrical activity in the thalamocortical network. “

*Alternatively the authors can revisit their model and represent more accurately the known effects of HA and ACh on each of the 'loose' cell types included in the model.*

As suggested by the reviewer, we examined a more detailed model to include additional effects of neuromodulators on the intrinsic currents that are specific to cell types. In these simulations, ACh influenced the K^+^-leak, M-current, H-current, persistent Na^+^ current and Ca^2+^-dependent K^+^ current in the cortical neurons. Histamine levels determined the K^+^-leak current and H-currents in all neuron types. The main qualitative predictions of the detailed model were similar to the reduced model. The four core model properties (K^+^-leak, Ih, GABA and AMPA) were critical to model transitions between sleep stages and selectively removing any of them, as shown in the Figure 5, led to the failure to explain sleep stage transitions as observed in vivo. Below we present the figure (now Figure 11) showing state transitions in a detailed model (Figure 11) and the failure of modeling transitions between sleep stages when the core properties were prevented to change in the detailed model (Figure 11).

We would like to note that the distributions of the intrinsic and synaptic currents in our model were varied across cell types. For instance, the cortical inhibitory neurons had lower conductance or absence of M-current, persistent Na^+^ and Ca^2+^-dependent K^+^ currents. In thalamic cells, intrinsic currents were different and were based on the data. Thus, the same neuromodulators influenced these different neuron types differently in our network model.

The following changes were implemented to the text:

We included Figure 11 which presents more detailed model of the effects of neuromodulators.

We also included the following text relevant to this figure in the Discussion section:

“We also tested a more detailed model, implementing the effects of ACh and HA on the other K^+^ currents as well as on the persistent Na^+^ and M-currents (Constanti and Galvan, 1983; McCormick and Williamson, 1989; Mittmann and Alzheimer, 1998). We observed that this detailed model qualitatively captured results similar to the minimal model (Figure 11). In contrast, after removing the changes of the K^+^-leak, Ih, GABA and AMPA currents but leaving the action of neuromodulators on the other channels, the detailed model failed to show transitions between sleep stages (Figure 11), suggesting that the cellular and synaptic properties identified in our study are critical in determining patterns of electrical activity in the thalamocortical network.”

*2) A similar concern applies to the handling of data gathered with ketamine anesthesia. The authors attempt to mimic ketamine in their model by decreasing NMDA's influence – but ketamine is a promiscuous molecule that has influence on a great number of channels. Additionally, the unconsciousness produced by the "dissociative" anesthetic, ketamine, has a very different phenotype than that of propofol or isoflurane which more or less produce a quiescence resembling sleep. As opposed to these GABAergic anesthetics, ketamine does not activate the sleep nuclei of the hypothalamus (VLPO), does not depress thalamic activity, and activates wake-promoting nuclei in the brain stem. The adjunct agent used in their* in vivo *experiments (xylazine) does on the other hand promote sleep, but does not do this through NMDA antagonism. Therefore the specificity of their model is really in question. George Mashour has done work on ketamine and his work might be better referenced.http://www.ncbi.nlm.nih.gov/pmc/articles/PMC4076669/*

*Further, we would expect that an increase in inhibition in the model would produce slow oscillations similar to sleep. This model state could be compared to* in vivo *data using propofol or isoflurane. Perhaps, this model is "tuned" to produce these stable oscillations in a variety of different parameter regimes – but as mentioned in the expert reviews (Reviewer #2), without an evaluation of the stimulation protocols it is difficult to determine what is a network effect vs. a reflection of the periodicity already present in the stimulation.*

We again thank the reviewer for this valuable comment. We choose to model Ketamine condition based on experimental evidences suggesting that ketamine inhibits NMDA receptors and ACh nicotinic and muscarinic receptors (Rudolph and Antkowiak, 2004; Alkire et al., 2008). However, we agree with the reviewer that ketamine could have many other effects, which are not fully understood. Thus, we replaced the section on the effect of ketamine with a simulation of the propofol anesthesia, as suggested by the reviewer.

Propofol increases decay time constant of the inhibitory post-synaptic potential (Kitamura et al., 2003). It is also known to reduce levels of ACh (Kikuchi et al., 1998) and inhibit TMN activity which is the source for HA (Nelson et al., 2002). In both animals and humans, propofol has shown to increases 8-15 Hz activity in the frontal regions and appearance of slow oscillations (Murphy et al., 2011; Purdon et al., 2013). In agreement with these previous studies, we present below an example of our in-vivo recordings from cat (this was a pilot experiment with N=1, so we decided not to include these data to the main text), that demonstrate increase in 8-15 Hz oscillations following the application of propofol (Figure 12). Note that the amplitude of 8-15 Hz oscillations in propofol condition was higher compared to natural sleep.

Based on these in vivo studies, we simulated effect of propofol by (a) increasing decay time constant of synaptic GABAergic inhibition as well as (b) modeling inhibition of ACh and HA release. Following such manipulations, we observed an increase in the 8-15 Hz activity (compare to our baseline model of the natural sleep). We then compared effect of increasing GABA time constant alone versus combined action of increasing GABA time constant and effect of reduction of ACh and HA. In the first case, we found no slow oscillations or oscillatory activity in 8-15 Hz. These findings suggest that propofol’s action may not be fully explained by its effect on synaptic inhibition and considering its indirect effect on release of other neuromodulator is required to observe the changes seen in vivo in the thalamocortical network.

Author response image 1.Multichannel recording from a cat during natural sleep (control) and propofol induced anesthesia condition.Top plot shows the recording electrode and picture of the implanted electrode. Middle plot shows the activity during natural sleep (raw data on top and filtered activity on bottom). Bottom plot show the activity during propofol anesthesia.**DOI:**
http://dx.doi.org/10.7554/eLife.18607.014

We now included the section “Modeling the effects of propofol anesthesia requires the combined action of ACh, HA and GABA”.

We have also replaced Figure 9.

In response to the reviewer comment about external stimulation to the model (please also see 2d reviewer comment below), we need to state that we have not been clear about nature of the stimulation in the model. We never applied an external persistent periodic or DC stimulation in any of the model simulations presented in the paper. All the model activity reported in this study results from intrinsic dynamics of the thalamocortical network. The only stimulation we applied, and what was mentioned in the original text, was a brief DC pulse to probe stability of the network. We have clarified this issue in the revised version of the manuscript.

To clarify the stimulation protocol, the following text was updated:

“During awake period, activity in the cortical PY and IN and thalamic RE cells was sparse, i.e., mean firing was around 2-3 Hz and 4-5 Hz in PY and IN cells, respectively, […] then the network returned to the baseline, as observed in experiments with a sensory input during alert awake conditions (Bruno and Sakmann, 2006; Hengen et al., 2016).”

And we described the stimulation in the Methods:

“Stimulation Protocol: A brief DC input lasting for 100 ms and amplitude of 1.5 nA was applied to TC neurons during awake state in some simulations to test stability of the network dynamics. The DC pulse was modeled using two Heaviside functions. No periodic stimulation was applied in any of the modeling experiments.”

*The expert reviews of reviewers #2 and #3 are appended to aid in the revision.*

*Reviewer #2:*

*[…]*

*Summary:*

*- Some parts of the summary are awkward in regard to sentence structure and punctuation. The second sentence does not sound like proper subject verb agreement, and the third sentence requires the word and between the two subjects to make sense. Α/theta bursts are mentioned in the Summary but not emphasized in the paper (previous version?) please remove.*

Thank you for pointing us to those issues. The summary section was edited to address reviewer’s critics and to clarify presentation. We also removed the mentioning of the α bursts in the summary. In the original manuscript, we still discuss α bursts when discussing the activity during REM sleep, however, it is scattered across the text and, we agree, that it is not significant enough to be mentioned in the summary.

*Introduction:*

*– First sentence of second paragraph should be reworded so characterize is not repeated. The first sentence of the third paragraph should start with: "Neuromodulators".*

We thank the reviewer for pointing out these changes. We revised the manuscript as it was suggested.

*Results*

– The TC stimulation is not represented in Figure 2. This input is also not described in the methods, given the strong excitatory connections between thalamus and cortex the nature of the stimulation ("simulating sensory input") could be driving the oscillations and must be included to evaluate the main conclusions of the paper.

Unfortunately, we missed to describe stimulation in sufficient details. We never applied an external persistent periodic or DC stimulation in any of the model simulations presented in the paper. All the model activity reported in this study results from intrinsic dynamics of the thalamocortical network. The TC stimulation, mentioned in the text, was applied as a brief DC pulse during awake stage to demonstrate the stability of the awake state activity. There was no external stimulation applied at any other time during the simulations. Below is the figure showing the response to brief TC stimulation.

We modified the text in the modified manuscript to clarify this further as shown below.

“During awake period, activity in the cortical PY and IN and thalamic RE cells was sparse, i.e., mean firing was around 2-3 Hz and 4-5 Hz in PY and IN cells, respectively, while the TC neurons primarily remained silent and RE cells were busting intermittently (Figure 2 and Figure 3). To test stability of the network dynamics, we applied thalamic stimulation in the form of a brief (100 msec) DC pulse once during awake state. It triggered transient increase of firing in cortical PY neurons and INs, and then the network returned to the baseline, as observed in experiments with a sensory input during alert awake conditions (Bruno and Sakmann, 2006; Hengen et al., 2016).”

We also added to the Discussion:

“All the network rhythms reported in this study result from intrinsic (autonomous) dynamics of the thalamocortical network; no external stimulation was applied to entrain any of the oscillatory activities in the model.”

And we described the stimulation in the Methods:

“Stimulation Protocol: A brief DC input lasting for 100 ms and amplitude of 1.5 nA was applied to TC neurons during awake state in some simulations to test stability of the network dynamics. The DC pulse was modeled using two Heaviside functions. No periodic stimulation was applied in any of the modeling experiments.”

*– Why did they examine such a large frequency range to evaluate phase-locking? When the connections are not completely known (*in vivo *recordings) phase-locking among narrow frequency bands (especially among specific regions, e.g., thalamus and cortex) is much more important for synchronization. Very little of the quantitative results rely on phase-locking.*

We thank the reviewer for point to this issue. It was a typo in our manuscript and the phase locking was computed for 0.5-20 Hz and not for 0.5-100 Hz. The only part of the result that utilized this phase locking analysis was for identifying the clusters of activity in Figure 4 and the 4B. Using a range from 0.5 to 20 Hz allowed for better detection of the clusters. We found that using very narrow power bands corresponding to slow oscillation (0.5-4Hz) or spindle activity (7-15 Hz) resulted in less clear distinction between the clusters and adding a dimension by using separate phase-locking value for δ and spindle bands did not improve the separation of clusters or a change in AIC curves. We now corrected the text to indicate the correct frequency range used for the phase locking.

– Why is ketamine compared to SWS? Others report it as sharing similarities with REM. Propofol or isoflurane anesthesia would be a better comparison.

*Methods*

We thank the reviewer for this important comment. Similar critique was also found in the reviewer 1 comments. We now removed the results on the ketamine and instead present results on the propofol anesthesia condition. We included new figure and a new section to discuss these new results. All the details about this section are presented in our response to question 2 of the “Essential revisions” (please see above).

*– Extrasynaptic and Intrasynaptic GABA concentrations are mentioned but not explored in the paper, please remove (unless added to discussion as to a potential limitation of the model).*

Indeed, in the Discussion section of the original manuscript we had a brief discussion of the Extrasynaptic vs Intrasynaptic GABA effects. Since the origin of change in GABA is not fully understood, we had to examine various possibilities whether the measured GABA levels reflect the extrasynaptic GABA concentration or the tonic GABA (Figure 10). As pointed by the reviewer, this is one of the limitations of the model due to the lack of clear experimental evidence on this topic. Therefore, we kept this text in the revised manuscript, however, we emphasized the reasoning for discussing this topic along with new supplementary figure. A detailed discussion on this topic was also provided to response to reviewer 3 (see below, last comment from reviewer 3). Particularly, the modified text was included in subsection “Role of GABA in sleep stage transitions.”

*– Network description: radius is not defined explicitly.*

Thank you. Similar comment was also made by Reviewer 1. We now modified the method section “Network structure” to include the details about network connectivity and radius.

*– The* in vivo *description mentions non-anesthetized cats and ketamine is given?*

We now removed the section on Ketamine (both animal data and model) and instead reported the modeling results on propofol induced anesthesia. Thus, in this revised version, all the cats recordings are from non-anesthetized animals.

*Discussion*

*– Much of the Discussion can be considerably shortened.*

We thank the reviewer for this suggestion. We removed several paragraphs from the Discussion section.

*- The Discussion does not adequately address other potential influences on spindle power (e.g., age and cortical volume).*

Indeed, many factors including age and cortical volume may impact the spindle and slow oscillation power. While the model is too simplified to address potential impact of the cortical volume, we included a brief discussion about possible changes with aging as shown below:

Furthermore, our results on the putative relationship between levels of neuromodulators and characteristic sleep electrical activities could be potentially relevant to understanding the changes observed with aging. Indeed, the level of the HA in cerebrospinal fluid is known to be elevated with aging (Prell et al., 1988), where spindle power is known to be reduced (Crowley et al., 2002). In our model, the HA level is one of the major predictors of spindle power. Thus, we can speculate that a reduction of HA with age could contribute to the decline of the spindle power.

*Reviewer #3:*

*This is a very interesting paper that uses a fairly detailed computational model, constrained by experimental observations in a variety of species, which is used to demonstrate the dependence of sleeps stage on a variety of network and intrinsic cell properties (leak current, AMPA, GABA, I(h)). The paper extends the findings to link sleep stages to two main neuromodulators known to be involved in controlling sleep – Ach and Histamine.*

*In general the paper is well written and the data beautifully presented. I very much liked the demonstration of different sleep stages with different intrinsic properties but it would have been good to refer to the very detailed model of Hill & Tononi which, a while ago, came to very similar conclusions.*

We greatly appreciate the reviewer’s positive comments about the manuscript. We thank the reviewer for recognizing the quality of our results and our approach. The reviewer’s critique has helped us make the manuscript more accessible to readers.

We thank the reviewer for pointing us about missing Hill et al. 2005 citation. Unfortunately, it disappeared by mistake after one of the internal revisions. We now cite this paper in the revised manuscript. In our new study we extend many of the early findings reported in Hill et al. 2005. Main advances of our new study include: (a) modeling all major sleep stages (including stage 2 and REM sleep) based on the biophysical mechanisms; (b) identifying role of the many other intrinsic and synaptic properties in sleep stage transitions (e.g., role of AMPA strength or Ih conductance strength); (c) reporting intermediate stages with spindle and slow oscillations that resulted from changing both ACh and HA; this was required to replicate the recordings from humans and animals.

We included this relevant citation in our discussion as follows:

“Previous computational studies proposed a minimal set of the mechanisms sufficient to model sleep spindles and slow oscillations (Destexhe et al., 1996; Bazhenov et al., 1998, 1999; Timofeev et al., 2000; Bazhenov et al., 2002; Compte et al., 2003b; Hill and Tononi, 2005; Bonjean et al., 2011). An increase in the K^+^ leak current was identified as a critical component for the transition between awake and slow wave sleep (Bazhenov et al., 2002; Hill and Tononi, 2005), and it was predicted that synchronization of the slow oscillation depends on cortico-cortical connections (Hill and Tononi, 2005). Several phenomenological and reduced mathematical models have been proposed to test the effects of neuromodulators on the sleep-wake cycle (McCarley and Hobson, 1975) (Robinson et al., 2011). Nevertheless, none of these previous models examined transitions among all major sleep stages based on biophysical mechanisms.”

*My only major problem with the manuscript comes from the attempt to relate the model parameter changes to histamine and Acetylcholine. I think a bit more detail and biological realism needs to be included here. For example, Ach is stated to affect AMPA-mediated transmission and leak current. Of the parameters used to construct the model cells it also affects Km and GABA release. Reading the methods it seems Ach was modelled as a scale factor for all intrinsic and synaptic properties – but this scale factor was increased for decreased Ach levels. This means, for example, that properties shown not to be affected by Ach were affected in the model, and properties of Ach well documented (such as depolarisation of main interneuron subtypes) were omitted or modelled conversely.*

We believe that we failed to present model equations properly in the Methods so it made an impression that”Ach was modelled as a scale factor for all intrinsic and synaptic properties”. Among all intrinsic currents ACh had only influenced the K^+^ leak current and had no effect on the other intrinsic currents. It also had effect on the synaptic AMPA-type transmission. We now modified our equations in Methods to clearly show the effect of ACh. Further, we would like to note that different cell types had different strength of intrinsic currents, including K^+^ leak, thus the same neuromodulator had varying effects on the different neuron types based on the cell specific combination of intrinsic currents.

The Methods section reads now:

“The cortical neuron model included dendritic and axo-somatic compartments, similar to the previous studies (Timofeev et al., 2000; Bazhenov et al., 2002); (Chen et al., 2012) and was described by the following equations,CmdVddt=−[AchPY.IdK−leak]−Idleak−IdNa−IdNap−IdCa−IdKCa−IdKm−Isymgcs(Vd−Vs)=−IsNa−IsK−IsNap

where the subscripts s and d correspond to axo-somatic and dendritic compartments, is the variable which is inversely proportional to the level of ACh in the cortex (AchPY=1 for awake state, AChPY= 1.25 for N2, AChPY=1.8 for N3 and =0.85 during REM) and determines the strength of the potassium leak current (IdK−leak)[…]”

We would like to add that, as also was suggested by the first reviewer, we now included the results from a detailed model (Figure 11), which implements the action of ACh and HA on the all K^+^ currents, Ih and persistent Na^+^ current. While these additional actions of ACh and HA were not sufficient by themselves to cause transitions between sleep stages, including these effects kept the qualitative features of the stage transitions that were reported in the minimal model (Figure 11).

*On a similar note, it is not clear how the 2 modulators chosen fit with the GABA levels used. There is a wealth of evidence linking Ach and Histamine to GABA release and postsynaptic effects – some which fit the basic hypothesis the authors propose (decreased Ach, increased GABA release), and some antagonistic to it (increased histamine, increased tonic GABA). This could be clarified with a more rigorous modelling of the known effects of the two agents to the conductances used in the model. If no changes in sleep stages are seen then this would provide evidence for an involvement of the other major neuromodulators involved in sleep not considered here.*

We again thank the reviewer for this important comment. Indeed, the origin of the changes in the GABA level during sleep is not well understood. In this study, we assumed that phasic GABA release is increased during stage 2 and 3 sleep, when ACh and HA are reduced. However, as suggested by the reviewer, there is a complex interaction between GABA and other neuromodulators. In our study, we examined in detail how varying levels of GABA influence the activity in the thalamocortical network for different combinations of ACh and HA (Figure 4). We identified that change in the level of GABA has relatively less effect compared to the changes in ACh and HA (note the clusters in Figure 4 do not change a lot in the z-axis as much as it changes in x and y-axis). However, for intermediate levels of ACh and HA, GABA determined the density of spindling activity (note the cluster 1,5 and 6 are located at the higher GABA levels). Further, we tested an alternative idea that experimental measurements of GABA reflect the extrasynaptic GABA (see figure below and Figure 10), which reflects amount of tonic inhibition. In these simulations, we found that changes in tonic inhibition did not alter the transition between stages but there were changes in amount of oscillatory activity in N2, N3 and REM stages. Due to the lack of clear experimental evidence on the origin of GABA changes across stages of vigilance, we were unable to make clear predictions about role of GABA in sleep stage transitions. Nevertheless, our results suggest that GABA through phasic and tonic inhibition alter the nature of activity within each sleep stage in the thalamocortical network. Furthermore, by examining the variations across wide range of GABA levels (eg in Figure 4) and in conditions of propofol anesthesia (Figure 9) our study could help guiding future experimental studies.

We now include text related to this issue in subsection “Role of GABA in sleep stage transitions.”